# Predicting mutational effects on protein binding from folding energy

**Arthur Deng** [1]  **Karsten Householder** [1]  **Fang Wu** [1]  **K. Christopher Garcia** [1]  **Brian Trippe** [1]

## Abstract

Accurate estimation of mutational effects on protein-protein binding energies is an open problem with applications in structural biology and therapeutic design. Several deep learning predictors for this task have been proposed, but, presumably due to the scarcity of binding data, these methods underperform computationally expensive estimates based on empirical force fields. In response, we propose a transfer-learning approach that leverages advances in protein sequence modeling and folding stability prediction for this task. The key idea is to parameterize the binding energy as the difference between the folding energy of the protein complex and the sum of the folding energies of its binding partners. We show that using a pre-trained inverse-folding model as a proxy for folding energy provides strong zero-shot performance, and can be fine-tuned with (1) copious folding energy measurements and (2) more limited binding energy measurements. The resulting predictor, STAB-DDG, is the first deep learning predictor to match the accuracy of the state-of-the-art empirical force-field method FoldX, while offering an over 1,000x speed-up.[1]

## 1. Introduction

Computation of mutational effects on binding energies is of central importance in structural biology and protein engineering. For example, three recent studies (Householder et al., 2024; Liu et al., 2024a; Johansen et al., 2024) designed proteins that bind to target proteins found on cancer cells, but the therapeutic promise of these molecules depends on their specificity; "off-target" binding to proteins differing at only one or two positions could induce toxicity to non-cancer cells (see Appendix A). In this setting and

others, accurate prediction of binding energies would support *in silico* design of proteins with requisite specificity. Given two interacting proteins and amino acid substitutions, the goal is to predict the differences in the change in Gibbs free energy upon binding (the "$\Delta\Delta G$" or "ddG").

Deep-learning (DL) methods have so far underperformed more classical methods based on empirical force fields in binding energy prediction. Notably, Bushuiev et al. (2024) find that recent results suggesting the superiority of DL predictors are confounded by dataset contamination — extant DL predictors generalize poorly when evaluated on interfaces not represented in the training set, and a Rosetta-based predictor Flex ddG provides state-of-the-art performance (Barlow et al., 2018; Bushuiev et al., 2024). Presumably, this underperformance of DL methods owes in part to the scarcity of training data, with experimental $\Delta\Delta G$ measurements for fewer than 350 distinct interfaces in the largest public curated dataset (Jankauskaitė et al., 2019).

In this work, we introduce a transfer learning approach that helps address this data limitation by reducing binding energy prediction to predicting folding energies. Our approach is based on two observations. First, as a consequence of the state function property of free energy, the binding energy between two proteins $A$ and $B$, $\Delta G_{\text{bind}}(A{:}B)$, can be computed as

$$\Delta G_{\text{bind}}(A{:}B) = \Delta G_{\text{fold}}(A{:}B) - \Delta G_{\text{fold}}(A) - \Delta G_{\text{fold}}(B), \tag{1}$$

where $\Delta G_{\text{fold}}$ denotes the free energy difference between *folded* and *unfolded* states of a protein monomer or complex (Figure 1a). This observation will allow us to use both folding stability and binding energy datasets in a supervised learning approach.

The second observation is that protein sequences and structure data in the Protein Data Bank (PDB) can inform an initial predictor of protein folding energies. This observation builds on the strong correlation between folding energies and likelihoods predicted by probabilistic models of protein sequences. Such correlations were first observed for Potts models (Lapedes et al., 2012; Hopf et al., 2017), and then later neural network models of sequence (Riesselman et al., 2018; Rives et al., 2021), as well as backbone structure conditional sequence models (Hsu et al., 2022a; Notin et al., 2023). We can leverage these correlations by using a pre-

---

[1]Stanford University. Correspondence to: Arthur Deng <lxdeng@stanford.edu>, Brian Trippe <btrippe@stanford.edu>.

*Proceedings of the 42nd International Conference on Machine Learning*, Vancouver, Canada. PMLR 267, 2025. Copyright 2025 by the author(s).

[1]Code: https://github.com/LDeng0205/StaB-ddG

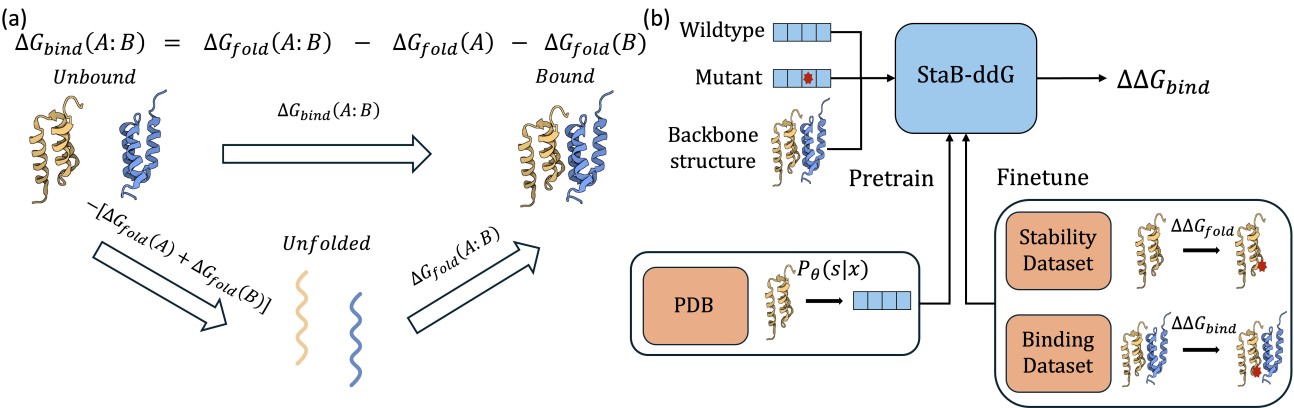

*Figure 1.* (a) Thermodynamic identity from the path independence of the free energy function (Equation (1)). (b) Schematic of STAB-DDG. STAB-DDG takes as input the backbone structure, a wild type sequence, and a mutant sequence to predict $\Delta\Delta G_{\text{bind}}$. STAB-DDG leverages three sources of data: structure/sequence pairs from the PDB, a folding stability dataset, and a binding affinity dataset.

trained sequence likelihood model as a zero-shot folding energy predictor.

Built on these observations, we present **Sta**bility-to-**B**inding **d**elta **d**elta **G** (STAB-DDG), a deep learning model to predict the mutation effects on protein-protein interaction energy using interface structure and sequence (Figure 1b). To fit STAB-DDG, we use a pre-trained inverse folding model ProteinMPNN (Dauparas et al., 2022) to initialize a zero-shot binding $\Delta\Delta G$ predictor according to Equation (1), and fine-tune on a combination of high-throughput folding energy measurements (Tsuboyama et al., 2023) and binding energy measurements (Jankauskaitė et al., 2019).

We demonstrate that fine-tuning STAB-DDG on a folding energy dataset improves binding energy predictions. Further fine-tuning on binding energy data provides state-of-the-art performance on a standard benchmark. Lastly, we evaluate STAB-DDG on two difficult case study datasets that corroborate these conclusions but illustrate that the problem remains difficult.

## 2. Preliminaries

In this section, we outline relevant background on protein thermodynamics and describe the notation used.

**Protein binding, folding, and mutational effects.** The binding energy between two proteins $A$ and $B$ is the free energy difference, $\Delta G_{\text{bind}}(A:B)$, between the *bound* and *unbound* (but folded) states of the system. Our goal is to predict the effect of mutations on binding energies. For a reference (wild type) interaction $A:B$ and mutant $A':B$, we write the mutational effect as

$$\Delta\Delta G_{\text{bind}}(A:B \to A':B) = \Delta G_{\text{bind}}(A':B) - \Delta G_{\text{bind}}(A:B).$$

Our starting point is the observation in Equation (1) that $\Delta G_{\text{bind}}$ may be computed as a difference in folding stabilities $\Delta G_{\text{fold}}$ as a consequence of the state function property of free energies; the change in energy along a direct path from the unbound to bound states is the same as the change along the path that proceeds through the unfolded state (Figure 1a).

Using Equation (1), we can express $\Delta\Delta G_{\text{bind}}$ in terms of folding energy as

$$\begin{aligned} \Delta\Delta G_{\text{bind}}(A:B \to A':B) =& \Delta\Delta G_{\text{fold}}(A:B \to A':B) \\ &- \Delta\Delta G_{\text{fold}}(A \to A'). \end{aligned} \tag{2}$$

Equation (2) underlies binding $\Delta\Delta G$ predictors based on empirical force fields (e.g., Guerois et al., 2002; Kastritis & Bonvin, 2013; Barlow et al., 2018). But to our knowledge this identity has not previously been leveraged in a DL binding $\Delta\Delta G$ predictor.

**Notation.** We represent a protein by its amino acid sequence $s = [s^1, s^2, ..., s^L]$ where $L$ is the length of the sequence and each $s^l \in \{1, \cdots, 20\}$ indicates the amino acid type. For binding partners $A$ and $B$, we write $s_{A:B} = (s_A, s_B)$ to denote the protein complex. Mutations of sequences $s$ are written as $s'$ and are assumed to be of the same length.

## 3. Predicting mutational effects on protein binding from folding energy

To incorporate labeled fine-tuning data with a pre-trained sequence likelihood model, we first propose STAB-DDG, a binding energy predictor based on sequence log probability. We show that STAB-DDG satisfies three properties we argue are desirable of $\Delta\Delta G$ predictors that are not satisfied

by previous predictors. Then, we present an objective to fine-tune STAB-DDG on both folding stability and binding affinity datasets. Lastly, we discuss variance reduction techniques to reduce prediction error at both training and inference time.

### 3.1. The STAB-DDG predictor

To obtain a binding $\Delta\Delta G$ predictor we start with a pre-trained sequence likelihood model to initialize a protein stability ($\Delta G_{\text{fold}}$) predictor as

$$f_\theta(s) = \log p_\theta(s), \tag{3}$$

where $p_\theta(s)$ is a probability model on sequences. We take the logarithm of $p_\theta$ to agree with the close-to-linear relationship between log probabilities of protein sequences and folding energies observed by Lapedes et al. (2012) and corroborated by many others (Notin et al., 2023). We use the ProteinMPNN inverse-folding model (Dauparas et al., 2022) for $f_\theta(s)$. ProteinMPNN depends additionally on a reference backbone structure, but we leave this dependence implicit to simplify notation.

Then, using Equation 1, we can obtain a binding affinity ($\Delta G_{\text{bind}}(A:B)$) predictor as

$$b_\theta(s_{A:B}) = f_\theta(s_{A:B}) - f_\theta(s_A) - f_\theta(s_B). \tag{4}$$

We refer to Equation (4) as the StaB parameterization because it links a **Sta**bility to **B**inding. Finally, we can use a difference of the predicted binding affinity between two complexes as a $\Delta\Delta G_{\text{bind}}$ predictor:

$$\Delta b_\theta(s, s') = b_\theta(s') - b_\theta(s). \tag{5}$$

We call the predictors with the form $\Delta b_\theta$ STAB-DDG predictors.

Computation of $\Delta b_\theta(s, s')$ involves computing $f_\theta$ on up to six systems; the complex and two binding partners for each of $s$ and $s'$. While in principle the backbone structures for each term could vary, we use backbone structures derived from a single complex for all 6 terms. This choice reflects an assumption that the backbone changes little upon binding and mutation.

**The choice of ProteinMPNN.** We choose ProteinMPNN to initialize $f_\theta$ and by extension $b_\theta$ and $\Delta b_\theta$. ProteinMPNN offers two advantages over sensible alternatives. First, compared to (even much larger) protein language models that do not take as input a reference backbone structure, Protein-MPNN provides stronger zero-shot folding stability predictions (Notin et al., 2023). ProteinMPNN's stronger performance presumably owes to the fact that mutational effects on binding are mediated through effects on structure.

The second advantage is that ProteinMPNN can make predictions for multi-chain complexes and multiple mutations. By contrast, most other folding stability predictors are implemented only for monomers and single mutations (Dieckhaus et al., 2024; Diaz et al., 2024). This complication, though likely surmountable with heuristics such as glycine linkers or residue gaps, is avoided with ProteinMPNN.

**Properties of the StaB-ddG predictor.** The form of $\Delta b_\theta$ constructed in Equations (3) to (5) imparts properties desirable of a $\Delta\Delta G_{\text{bind}}$ predictor. We formalize these properties in the following proposition.

**Proposition 3.1.** *Consider the class of binding energy predictors $\mathcal{B} = \{\Delta b_\theta\}$, with $\Delta b_\theta$ parameterized as in Equation (5) by $p_\theta(s)$ that is an arbitrary $20^L$-simplex valued function of $s$. The family of predictors $\mathcal{B}$ satisfies*

1. *Antisymmetry: for any $\Delta b_\theta$ in $\mathcal{B}$,*

   $$\Delta b_\theta(s, s') = -\Delta b_\theta(s', s),$$

2. *Mutational path independence: for any $\Delta b_\theta$ in $\mathcal{B}$ and $s, s', s''$,*

   $$\Delta b_\theta(s, s') = \Delta b_\theta(s, s'') + \Delta b_\theta(s'', s'), and$$

3. *Expressivity (Informal): for any dataset of binding free energy measurements, there exists a $\Delta b_\theta$ in $\mathcal{B}$ that fits the measurements exactly.*

*Proof: Properties 1 and 2 follow immediately from the construction of $\Delta b_\theta$ as the difference of evaluations of $b_\theta$ defined in Equation (5). Appendix B provides a formal statement of the expressivity property along with a proof.*

Because $\Delta\Delta G_{\text{bind}}$'s are differences by definition they satisfy Properties 1 and 2 of Proposition 3.1. Though these properties are readily obtained in our predictor by construction, they are nonetheless not satisfied by other recent DL predictors (Table 1).

Property 3 requires $p_\theta(s)$ to be able to take arbitrary values on the simplex. In practice, $p_\theta(\cdot)$ is parametrized by ProteinMPNN which, as an auto-regressive model parameterized by a deep message-passing neural network, can approximate to arbitrary simplex-valued functions. This property formalizes the ability of our predictor to model epistasis and achieve zero training loss on the fine-tuning dataset. In contrast, a predictor parameterized by a masked language model (e.g., Bushuiev et al., 2024) cannot model non-additive effects between multiple mutations (Appendix B), and force field-based methods (e.g., FoldX) do not have this property.

*Table 1.* Thermodynamic properties of different $\Delta\Delta G$ predictors. See Appendix B for details.

| Predictor | Anti-symmetry | Mut. Path Independence | Express-ivity |
|---|---|---|---|
| FoldX | $\checkmark$ | $\checkmark$ | $\times$ |
| Flex ddG | $\checkmark$ | $\checkmark$ | $\times$ |
| Surface-VQMAE | $\times$ | $\times$ | $\checkmark$ |
| Prompt-DDG | $\times$ | $\times$ | $\checkmark$ |
| DiffAffinity | $\times$ | $\times$ | $\checkmark$ |
| ProMIM | $\times$ | $\times$ | $\checkmark$ |
| RDE-Net | $\checkmark$ | $\times$ | $\checkmark$ |
| PPIformer | $\checkmark$ | $\times$ | $\times$ |
| **STAB-DDG** | $\checkmark$ | $\checkmark$ | $\checkmark$ |

### 3.2. Assimilation of folding and binding energy data

Though our goal is to predict binding, the number of binding energy measurements available in the largest public curated set is two orders of magnitude fewer than that in the largest comparable set of folding stability measurements (Table 2). As such, we adopt a sequential fine-tuning strategy, where we first fine-tune on folding stability data and then fine-tune on more limited binding affinity data.

*Table 2.* Dataset size comparison between PDB and the largest available stability and binding datasets.

| DATASET | # OF STRUCTURES | # OF MEASURED $\Delta\Delta G$ |
|---|---|---|
| PDB | 230,744 | — |
| STABILITY | 412 | 776,298 |
| BINDING | 345 | 7,085 |

**Fine-tuning to folding stability data.** The Megascale stability dataset is the largest publicly available dataset on protein folding energy, with 776,298 folding stability measurements across 412 small monomeric proteins from a high-throughput assay (Tsuboyama et al., 2023). We follow the same dataset preparation protocol as described by Dieckhaus et al. (2024), but keep entries with multiple mutations. We represent the Megascale stability dataset with $N$ structures and $M_n$ mutants associated with the $n$th structure as

$$\mathcal{D}_{\text{fold}} = \{(x_n, s_{n,\text{ref}}, y_{n,\text{ref}}, \{s_{n,m}, y_{n,m}\}_{m=1}^{M_n})\}_{n=1}^N,$$

where $s_{n,\text{ref}}$ and $y_{n,\text{ref}}$ denote the reference sequence and $\Delta G$, and $x_n$ is a predicted reference structure. We use $\{s_{n,m}, y_{n,m}\}_{m=1}^{M_n}$ to denote the set of mutant sequences and corresponding mutant $\Delta G$ values. A set of $\Delta\Delta G$ values can then be computed by taking the difference between mutant and reference $\Delta G$.

To fine-tune $\theta$ on $\mathcal{D}_{\text{fold}}$, we construct a $\Delta\Delta G_{\text{fold}}$ predictor as

$$\Delta f_\theta(s, s') = f_\theta(s') - f_\theta(s), \tag{6}$$

where we use the same structure $x_n$ to compute $f_\theta(s')$ and $f_\theta(s)$. Then, we fine-tune by minimizing

$$L_{\text{fold}}(\theta, \mathcal{D}_{\text{fold}}) = \frac{1}{N} \sum_n^N \frac{1}{M_n} \sum_m^{M_n} (\Delta f_\theta(s_{n,\text{ref}}, s_{n,m})$$
$$- (y_{n,m} - y_{n,\text{ref}}))^2, \tag{7}$$

where the $\frac{1}{M_n}$ scaling ensures that each complex has equal contribution to the loss.

**Fine-tuning to binding affinity data.** We use SKEMPIv2.0, the largest publicly available binding affinity dataset with 7,085 binding $\Delta\Delta G$ measurements across 345 complexes, for fine-tuning STAB-DDG and comparing it against other baseline methods (Jankauskaitė et al., 2019). SKEMPIv2.0 contains errors from the manual curation process, such as mislabelled entries or entries with different $\Delta\Delta G$ values for the same mutation. Here, we apply a filtering procedure to the dataset based on one applied to SKEMPIv1.0 from previous work (Dourado & Flores, 2014; Barlow et al., 2018). Further, conducting comparisons on SKEMPIv2.0 fairly requires careful consideration. Bushuiev et al. (2024) pointed out data leakage based on homology in previous train/test splits of the dataset. However, the held-out test set used by Bushuiev et al. (2024) only contained five interface clusters. To address these problems, we divide the dataset based on the annotated structurally homologous clusters and apply a random train/test split, with 121 complexes in the fine-tuning dataset and 80 complexes in the held-out test split. The filtering and splitting procedure is fully described in Appendix C.

Analogously to the Megascale stability dataset, SKEMPIv2.0 can be instantiated as

$$\mathcal{D}_{\text{bind}} = \{(x_n, s_{n,\text{ref}}, y_{n,\text{ref}}, \{s_{n,m}, y_{n,m}\}_{m=1}^{M_n})\}_{n=1}^N$$

with the difference being $y_{\text{ref},n}$ and $y_{n,m}$ referring to binding $\Delta G$ instead of folding $\Delta G$ and $x_n$ representing crystal structures instead of predicted structures. We fine-tune on these data by minimizing

$$L_{\text{fold}}(\theta, \mathcal{D}_{\text{bind}}) = \frac{1}{N} \sum_n^N \frac{1}{M_n} \sum_m^{M_n} (\Delta b_\theta(s_{n,\text{ref}}, s_{n,m})$$
$$- (y_{n,m} - y_{n,\text{ref}}))^2. \tag{8}$$

### 3.3. Variance reduction by Monte Carlo ensembling and antithetic variates

The choice to use ProteinMPNN as our parameterization of $f_\theta(s)$ introduces model-specific stochasticity in the form of

a randomized decoding order and Gaussian noise to backbone coordinates. This stochasticity introduces variance that contributes to the prediction error.

We make explicit the dependence of the model output on the stochasticity as $b_\theta(s|\epsilon)$ for a random variable $\epsilon$. Then, for two sequences $s$ and $s'$, and a measurement $y = \Delta\Delta G_{\text{bind}}$, we can decompose the expected prediction error into contributions from squared bias and variance (see e.g., Hastie et al., 2009, Chapter 7) as

$$
\mathbb{E}[(b_\theta(s'|\epsilon') - b_\theta(s|\epsilon) - y)^2] = \\
\underbrace{(\mathbb{E}[b_\theta(s'|\epsilon') - b_\theta(s|\epsilon)] - y)^2}_{\text{Bias}} + \underbrace{\text{Var}[b_\theta(s'|\epsilon') - b_\theta(s|\epsilon)]}_{\text{Variance}},
$$

where the randomness is taken over the stochasticity $\epsilon$ and $\epsilon'$. We reduce the variance in two ways.

**Antithetic variates.** The first way is an instance of the antithetic variates method (Hammersley & Morton, 1956). The key idea is that the decomposition

$$
\text{Var}[b_\theta(s'|\epsilon') - b_\theta(s|\epsilon)] = \text{Var}[b_\theta(s'|\epsilon')] + \text{Var}[b_\theta(s|\epsilon)] \\
- 2\text{Cov}[b_\theta(s|\epsilon), b_\theta(s'|\epsilon')]
$$

reveals that the correlation of $b_\theta(s|\epsilon)$ with $b_\theta(s'|\epsilon')$ decreases the overall variance. So any coupling of $\epsilon$ and $\epsilon'$ for which $\text{Cov}[b_\theta(s|\epsilon), b_\theta(s'|\epsilon')]$ is positive will lead to lower variance than if $\epsilon$ and $\epsilon'$ were sampled independently. We accomplish this by fixing $\epsilon' = \epsilon$, which we implement by using the same permutation order and backbone noise for the wild type and mutant systems for each $\Delta\Delta G$ prediction.

**Monte Carlo averaging.** The second way is Monte Carlo averaging. By replacing each prediction with its average across $M$ independently sampled permutation orders and backbone noise samples, the variance is reduced by a factor of $M$. Ensembling can be applied together with the antithetic variates method by fixing $\epsilon' = \epsilon$. Note that ensembling over more samples increases the compute cost. We discuss the effects of ensemble size in Section 5.

## 4. Related work on predicting mutational effects on binding affinity

Existing approaches for predicting mutation effects on binding $\Delta\Delta G$ can be categorized as empirical force field-based methods and DL-based. Force field-based methods use energy functions to model inter-atomic interactions (Guerois et al., 2002; Park et al., 2016; Barlow et al., 2018; Sampson et al., 2024). While these methods have long dominated the field, they are often computationally expensive and have limited accuracy. For example, Flex ddG (Barlow et al., 2018) — a predictor based on Rosetta (Alford et al., 2017) — requires multiple CPU-hours per mutation but typically

produces estimates with Pearson correlation to experimental $\Delta\Delta G$s no larger than $R \approx 0.65$ (Barlow et al., 2018).

Free-energy perturbation (FEP) defines a class of potentially more accurate methods for estimating mutational effects (Zwanzig, 1954). Recent studies using FEP (Sergeeva et al., 2023) demonstrate small improvements over Flex ddG and related methods. However, these studies rely on a closed-source software implementation and case-specific expert tuning (see e.g. Sampson et al., 2024), and are even more computationally expensive. Consequently, we are unable to assess the accuracy of FEP methods.

Much recent work on $\Delta\Delta G$ prediction methodology has focused DL approaches. Several prior works on DL methods (Luo et al., 2023; Liu et al., 2024b; Mo et al., 2024; Bushuiev et al., 2024; Wu & Li, 2024; Wu et al., 2024) have claimed to deliver performance surpassing force field-based predictors (e.g., FoldX and Flex ddG) based on performance on the SKEMPIv2.0 (Jankauskaitė et al., 2019) benchmark dataset. However, Bushuiev et al. (2024) find that the train/test splits used to support these claims suffer from data leakage; once this data leakage is corrected the performance of these deep learning predictors lag Flex ddG.

Jiao et al. (2024) decompose $\Delta\Delta G$ computation into mutational effects on bound and unbound states. The resulting predictor has the same form as STAB-DDG (though without variance reduction), but the authors do not provide an interpretation of the in terms of folding energy that enables fitting to folding stability data. Frellsen et al. (2025) study the relationship between inverse folding models and folding energies and suggest this interpretation of the results of Jiao et al. (2024), but their empirical studies do not consider binding energies.

Gong et al. (2023) proposes an approach that leverages a pre-trained folding stability model, but does not leverage the thermodynamic identity to parameterize binding energies as folding energies.

Other recent work considers zero-shot $\Delta\Delta G$ predictions from 3D structure models (e.g. Jin et al., 2023), but does not achieve performance comparable to FoldX (Appendix E.3).

## 5. Experiments

To evaluate STAB-DDG, we first analyze the contributions of different techniques that lead to an improvement in "zero-shot" $\Delta\Delta G_{\text{bind}}$ prediction accuracy, without training on $\Delta\Delta G_{bind}$ data. Next, we introduce baseline methods and show that STAB-DDG is the only DL approach to match FoldX and Flex ddG; an ensemble constructed by averaging FoldX and STAB-DDG provides state-of-the-art performance. Finally, we evaluate out-of-distribution accuracy of our approach on two additional binding strength datasets:

one consisting of *de novo* designed small protein binders, and a second consisting of T cell receptor (TCR) mimic proteins we curate.

In protein engineering applications, a $\Delta\Delta G_{\text{bind}}$ prediction may be used to rank candidate sequence variants of an interface of interest to select as a subset for experimental screening. Therefore, we compute Spearman's rank correlation coefficient for mutational effects and predictions for each interface, and report the mean of this metric across complexes, along with standard errors. We refer to this metric as "per interface Spearman". When we compute per interface Spearman, we consider only complexes with 10 or more mutants; below this threshold, this metric suffers from high variance (Appendix C).

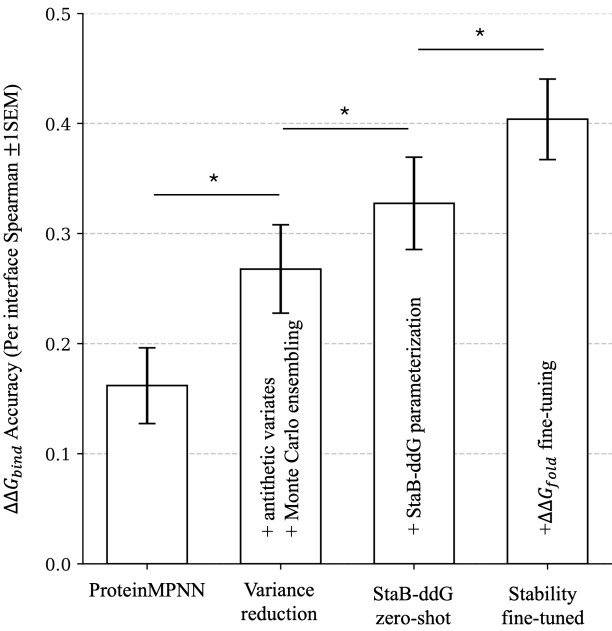

*Figure 2.* Evaluation of zero-shot binding predictors on the binding data training split. ProteinMPNN refers to using log-likelihoods of entire complexes ($\Delta f_\theta$) from the pre-trained ProteinMPNN weights. Asterisks (*) denote significance (one-sided paired t-test) at $p<0.05$.

## 5.1. Contributions to zero-shot $\Delta\Delta G_{\text{bind}}$ accuracy.

We first examined the individual contributions of techniques from our method that, starting from ProteinMPNN, led to a zero-shot binding energy predictor that incorporates information from folding stability data (Figure 2). We evaluated the binding energy prediction accuracy of different zero-shot predictors on the binding data training split described in Section 3.2.

**Variance reduction.** To reduce the error from the stochasticity inherent to ProteinMPNN, we applied the variance

reduction techniques described in Section 3.3 and observed improved accuracy. Specifically, reducing the variance of the ProteinMPNN predictor by (1) fixing the decoding order and backbone noise between the wild type and mutant sequences, and (2) ensembling over 20 predictions significantly improved zero-shot performance (Figure 2). We found that fixing the decoding order and backbone noise also led to better training dynamics, and provided empirical validation for the choice of ensemble size in Appendix D.

**STAB-DDG zero-shot.** We applied the pre-trained weights with variance reduction in the form of the binding predictor $\Delta b_\theta(s, s')$. The resulting predictor, STAB-DDG zero-shot, uses the same weights as ProteinMPNN but achieved significantly better accuracy (Figure 2).

**Fine-tuning on folding stability data.** To test whether fine-tuning on additional folding stability data translates to improved binding energy prediction accuracy, we fine-tuned STAB-DDG zero-shot on the folding stability dataset ("Stability fine-tuned"). We found that including these data further increased binding prediction accuracy (Figure 2).

We next wondered whether the folding stability data were sufficient to remove the need for unsupervised pre-training. We found that training directly on the folding stability dataset without inverse folding pre-training led to significantly worse binding prediction accuracy (Appendix D).

We validated our approach of fine-tuning on folding stability data of Tsuboyama et al. (2023) by comparing to a state-of-the-art folding stability predictor ThermoMPNN (Dieckhaus et al., 2024). ThermoMPNN is also based on ProteinMPNN but adds a transfer-learning module to output predictions. We found that despite not introducing additional parameters to ProteinMPNN, our stability fine-tuned model achieved performance not much lower than ThermoMPNN; our predictor provided a Spearman (over all domains) of 0.69 vs. 0.73 for ThermoMPNN (Appendix E).

We additionally explored different forms of the predictor and training techniques that did not have a sizeable effect. First, we tried fitting amino-acid-specific offsets to the predictor in the form of a linear model to correct for the initial scale mismatch between sequence log-likelihoods and free energy, measured in kilocalories per mole. However, adding these terms did not have a significant effect on binding prediction accuracy (Appendix D). Second, we experimented with using AlphaFold3 (Abramson et al., 2024) predicted structures to more accurately model the unbound (apo) structures of individual binders, instead of using the structure of the bound conformation. While using predicted apo structures improved several other metrics, this modification did not improve the per interface Spearman (Appendix D).

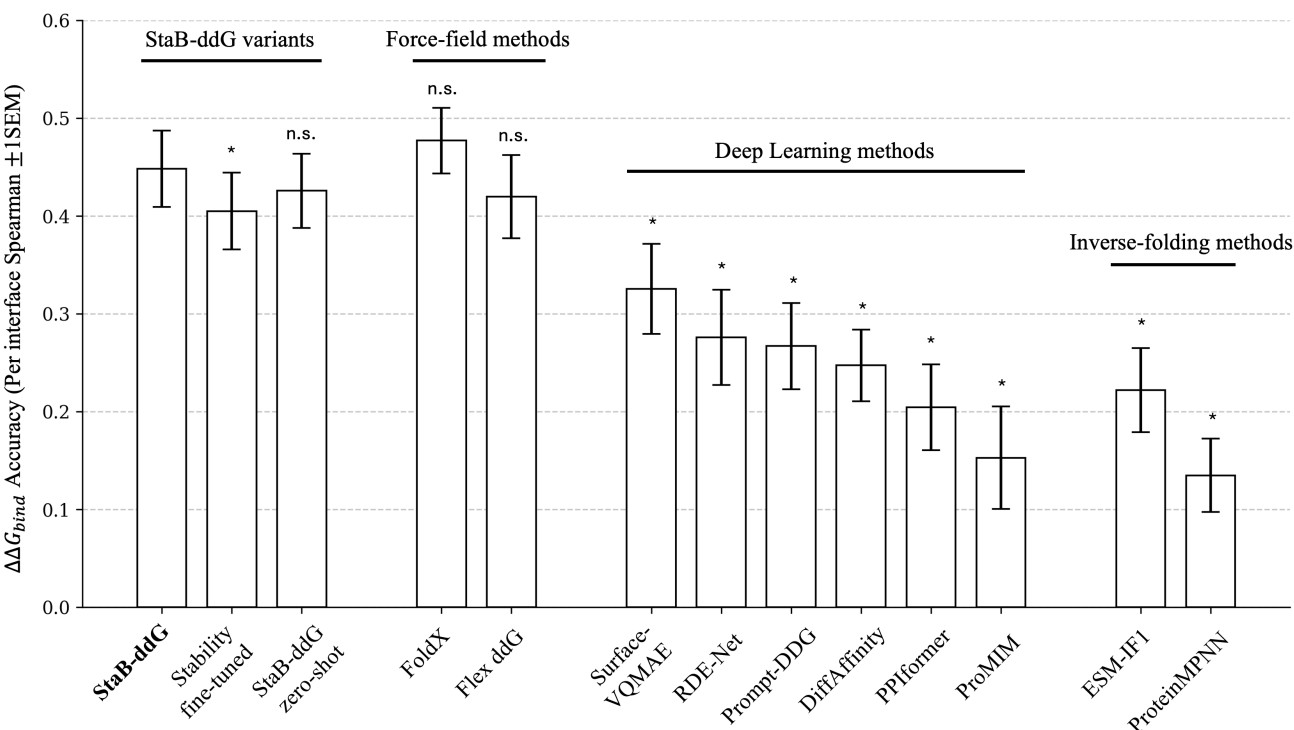

*Figure 3.* Evaluation of accuracy on the binding $\Delta\Delta G_{\text{bind}}$ benchmark test split of SKEMPIv2. Left: STAB-DDG and its variations. Middle: Previous deep learning methods. Right: Inverse Folding models. *: significance (two-sided paired t-test with STAB-DDG) at $p<0.05$, n.s.: not significant.

## 5.2. Comparison to existing methods

Using Stability fine-tuned as a starting point, we further fine-tuned this model on the binding data train split described in Section 3.2. We call the resulting predictor STAB-DDG.

We compared STAB-DDG to baseline methods on the binding data test split. Figure 3 presents per interface Spearman. We find similar trends for other metrics considered in previous works (Appendix E). We describe the baselines and then the results.

**Baselines.** We compare StaB-ddG to 10 baseline predictors. These comprise methods based on empirical force fields, supervised deep learning methods, and "zero-shot" inverse-folding methods.

FoldX (Guerois et al., 2002) and Flex ddG (Barlow et al., 2018) provide $\Delta\Delta G_{\text{bind}}$ predictions based on approximate energy functions. For both predictors, the parameters of the energy functions are fit to agree with empirical data; for FoldX weights on the components of a bespoke energy function are fit by a grid search and for Flex ddG nonlinear scalings of the component terms of the Rosetta energy function (Alford et al., 2017) are fit using a generalized additive model (Hastie & Tibshirani, 1986).

Both FoldX and Flex ddG have hyperparameters that must be chosen to make a prediction. We found that the choices of these parameters have a significant impact on the quality of their predictions. For Flex ddG, Barlow et al. (2018) shows that using many backbone conformations generated by "backrub" sampling significantly increases accuracy and (in their public code) recommends using the average prediction across several runs; we use Rosetta version 3.8 with 35,000 backrub steps and average predictions across 10 models. For FoldX, a variable number of `Repair` steps can be made before predictions. These steps make small changes to the coordinates of input crystal structures to which the energy function is extremely sensitive, and without which we found $\Delta\Delta G_{\text{bind}}$ bind predictions to be meaningless. Following Sergeeva et al. (2020), we perform 5 `Repair` steps before scoring. We use FoldX version 4.1.

We suspect that suboptimal choices of the parameters of these methods have led their accuracy to be underestimated by recent papers proposing deep learning $\Delta\Delta G_{\text{bind}}$ prediction methods. For example, (Bushuiev et al., 2024) runs Flex ddG without backrub steps (presumably due to compute cost) and (Jin et al., 2023) runs FoldX with only one `Repair` step.

We compared to six supervised DL baselines: Surface-

VQMAE (Wu & Li, 2024), RDE-Net (Luo et al., 2023), Prompt-DDG (Wu et al., 2024), DiffAffinity (Liu et al., 2024b), PPIformer (Bushuiev et al., 2024), and ProMIM (Mo et al., 2024). Each of these methods involves first pre-training on protein sequence or structure data and then fine-tuning on binding energy data. We repeat the fine-tuning stage of each method using our train/test split. We did not compare to Boltzmann Alignment (Jiao et al., 2024) as the parametric form of the predictor is similar to STAB-DDG zero-shot.

We included inverse-folding models ESM-IF1 and ProteinMPNN as unsupervised baselines (Dauparas et al., 2022; Hsu et al., 2022b). Zero-shot predictions were computed by subtracting the wild type complex sequence log-likelihood from the mutant log-likelihood.

**Results.** When evaluated on the binding data test split, STAB-DDG achieved higher per interface Spearman (0.45), outperforming previous DL methods (Figure 3 and Table 7). Similar to Bushuiev et al. (2024), we found that the DL baselines underperform FoldX and Flex ddG when evaluated on an interface homology-based split. We found STAB-DDG zero-shot to be surprisingly competitive, also outperforming previous DL methods, despite using the same model weights as ProteinMPNN. However, we did not find the difference between STAB-DDG, STAB-DDG zero-shot, FoldX, and Flex ddG to be statistically significant based on a two-sided t-test.

In contrast to the result on the training split, we observe lower accuracy for the stability fine-tuned model as compared to StaB-ddG zero shot on our test split. We attribute this inversion to the small number of clusters in the test split.

To obtain a state-of-the-art predictive model, we found that averaging the predictions from STAB-DDG and FoldX achieved a higher accuracy than any previous method (Per Interface Spearman 0.53) (Appendix E).

**Stratification of performance.** To understand how STAB-DDG performs on different types of complexes, we performed an analysis on different subsets of the binding data test set. We stratified the test set according to interface rigidity and complex size (Table 3). As a proxy for interface rigidity, we computed the loop content at the interface; specifically, we considered all residues within 10 Å of an atom in another chain and computed the fraction of these residues with secondary structure annotated as loop. We found that STAB-DDG performed better for more rigid, smaller complexes.

The better performance on rigid interfaces may be explained by STAB-DDG's use of a single static structure for prediction of folding energy for both the complex and monomers; flexible interfaces are more likely to change upon binding

*Table 3.* Performance of STAB-DDG across stratification of binding test set; Root Mean Squared Error (RMSE) in Kcal/mol for different interface loop content and complex size (number of residues) thresholds.

| Loop Content | RMSE | Complex Size | RMSE |
|---|---|---|---|
| < 30% | 1.12 | < 150 | 0.94 |
| < 40% | 1.29 | < 200 | 0.99 |
| < 50% | 1.46 | < 400 | 1.28 |
| < 60% | 1.50 | < 600 | 1.42 |
| < 70% | 1.48 | < 800 | 1.38 |
| < 80% | 1.48 | < 1000 | 1.50 |

and will be less well represented by a single structure.

The better performance on smaller complexes may be explained by the bias in the composition of the folding stability data of Tsuboyama et al. (2023), which consists only of small <80 residue domains.

**Experimental details.** In summary, we fine-tuned on the Megascale stability dataset using the ADAM optimizer with a learning rate of 3e-5 for 70 epochs with a batch size of 25,000 amino acids. We fine-tuned on SKEMPIv2.0 using the ADAM optimizer with learning rate 1e-6 for 200 epochs with a batch size of 25,000 amino acids. We provide training and inference code at `https://github.com/LDeng0205/StaB-ddG`.

**Running time.** STAB-DDG is orders of magnitude faster than Flex ddG and FoldX by benefiting from GPU parallelism. Flex ddG is the most computationally expensive of the three, requiring roughly 15 CPU hours per mutation. For FoldX, initial "repair" steps are computed on the wild-type interface PDB followed by scoring of individual mutants. On our filtered SKEMPI binding dataset, the total compute time was roughly 260 CPU hours for 4451 mutants (210 seconds per mutation). For StaB-ddG, by contrast, predictions on the same dataset took 13 NVIDIA-5090 GPU-minutes with batched computation (0.2 seconds per mutation). This runtime corresponds to a $1000\times$ speedup over FoldX on a single device.

Model finetuning of STAB-DDG took 10 hours and 5 hours on the Megascale stability dataset and the SKEMPIv2.0 training split on a single H100 GPU.

## 5.3. Generalization of prediction performance on two case study datasets.

Despite STAB-DDG having achieved state-of-the-art performance on SKEMPIv2.0, the statistical power of the conclusions drawn was limited by the size of the dataset and experimental noise. Thus, it remains to be validated whether our conclusions still hold and if current computational bind-

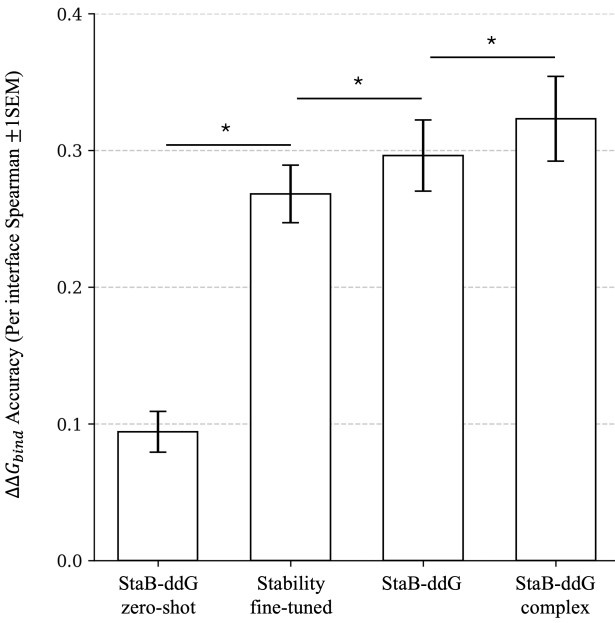

*Figure 4.* Comparison of STAB-DDG zero-shot, Stability fine-tuned model, and STAB-DDG on the yeast surface display dataset. * denotes significance (one-sided paired t-test) at $p<0.05$.

ing $\Delta\Delta G$ prediction tools are readily useful in settings not represented in SKEMPIv2.0. We sought to address this problem by evaluating STAB-DDG in two case studies, de novo designed "mini-binder" yeast surface display binding dataset and a curated TCR mimic dataset.

**Yeast surface display case study.** To validate the effects of fine-tuning on folding stability data and further fine-tuning on binding data, we compared STAB-DDG zero-shot, Stability fine-tuned, and STAB-DDG on site saturation mutagenesis data from Cao et al. (2022), which has been used to perform retrospective evaluation of DL-based binder design methods (Bennett et al., 2023). The dataset contains sequence count information from yeast surface display libraries of 28,293 single mutants across 33 interfaces of Rosetta-designed small protein binders with various natural targets. The sequence counts are then used to estimate a proxy for the dissociation constants of each binder, which we relate to a $\Delta\Delta G$ estimate (Appendix E).

We found that both fine-tuning on folding stability and binding affinity improved binding prediction accuracy (Figure 4). However, we found that the folding energy predictor for the entire complex achieved a higher per interface Spearman than our binding predictor parameterization. Appendix E demonstrates that this observation is explained through the experimental readout confounding expression levels with binding energy in the yeast-display based assay. In brief, the binding energy proxy of a particular variant depends on both its binding affinity and its expression, a quantity closely related to folding stability.

**TCRm case study.** We curated a set of 30 $\Delta\Delta G$ measurements from six TCR mimic antibody structures determined by surface plasmon resonance (SPR) by searching through all TCR mimic structures in the TCR3d database (Appendix A) (Gowthaman & Pierce, 2019). We next evaluated STAB-DDG on these data and found an overall Spearman correlation of $0.13 \pm 0.39$ (with standard error computed with a cluster bootstrap, see Appendix C.2), and so are unable to conclude whether or not STAB-DDG is predictive in this setting. This negative result is likely due to a combination of a small sample size and the greater difficulty of predicting $\Delta\Delta G_{\text{bind}}$ for loopy antibody residues.

## 6. Discussion

Accurate computational prediction of the mutational effects on protein interaction binding energies could significantly improve the potency and specificity of protein therapeutics. Despite years of interest in improving such predictions with deep learning, success has been minimal. By achieving performance comparable to FoldX, STAB-DDG marks an important step in this direction.

Computational prediction of mutational effects on binding energies remains a challenging open problem. Several areas remain to be explored. First, STAB-DDG does not model changes in the backbone upon a mutation; in general, mutations on flexible regions of the binding site are likely to introduce fluctuations in the backbone structure. Second, while the capacity to improve StaB-ddG by including larger and more diverse binding and folding energy datasets presents an advantage relative to empirical force-field-based predictors, the rate of improvement as a function of these ingredients and how best to assimilate these data into a single predictor requires further investigation.

## Acknowledgements

We thank Sebastian Thrun for providing valuable high level guidance on project direction. We thank Henry Dieckhaus, Brian Kuhlman, and Gabriel Rocklin for helpful early discussions, and Christopher Fifty, James Bowden, and Henry Smith for helpful comments that improved the manuscript.

## Impact Statement

This paper presents work whose goal is to advance the machine learning methods for modeling and design of proteins. There are many potential societal consequences of our work, none which we feel must be specifically highlighted.

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

# Appendix Table of Contents

# A. Additional background on binding $\triangle\triangle G$ prediction and TCR mimic engineering

## A.1. Background

**Binding specificity.** The binding specificity of a protein can be characterized by the difference in binding affinity between the reference, or wild-type, interaction and off-target interactions, where binding affinity is the free energy difference ($\triangle G$) between the bound and unbound states of a system of two proteins. More concretely, a binding affinity difference of $\triangle\triangle G$ = 1 *kcal/mol* between a cancer target and its healthy analogue would translate to 10X higher binding affinity for the cancer target. Thus, binding specificity can be expressed by a series of $\triangle\triangle G$ values between the wild type interaction and a known list of off-targets.

**TCR mimic specificity.** TCR mimic antibodies hold significant promise for cancer-specific immunotherapy (Klebanoff et al., 2023; Yang et al., 2023). These engineered molecules are designed to selectively bind to cancer-associated peptides presented on major histocompatibility complexes (pMHCs) while avoiding recognition of off-target peptides displayed on healthy cells. Given that these peptides are typically only 9–12 amino acids long, the challenge lies in distinguishing cancer-associated pMHCs from normal pMHCs, which can sometimes differ by just a single amino acid (Rossjohn et al., 2015). Achieving this level of specificity is critical, as even minor cross-reactivity could lead to severe dose-limiting toxicities or fatal depletion of essential healthy cells (Linette et al., 2013). Predicting off-target toxicity is particularly difficult because the potential peptide landscape is vast—ranging from approximately $20^9$ to $20^{12}$ theoretical peptide combinations. As a result, experimental screening alone is often insufficient to fully assess specificity (Birnbaum et al., 2014; Holland et al., 2020). Computational approaches that refine TCR mimic binding to maximize selectivity could significantly reduce toxicity risks while enhancing precision and the molecule's therapeutic window. By improving specificity, such strategies could accelerate the development of safer and more effective TCR mimic therapies, ultimately broadening their clinical utility.

## A.2. TCR mimic case study

To curate this case study, we searched the TCR3d Database for structures of TCR mimic antibodies bound to pMHC that were deposited in the PDB and had associated surface plasmon resonance (SPR) data with mutations (Gowthaman & Pierce, 2019). We prioritized SPR data because it provides the most quantitatively accurate and sensitive measurements of binding affinity changes, making it a reliable source of binding $\triangle\triangle G$. TCR mimic antibodies contain flexible loops with many degrees of freedom, making the effects of mutations on affinity and specificity particularly difficult to predict. In total, we identified six TCR mimic complexes with available mutational and SPR data from the literature (PDB IDs: 3HAE, 6UJ9, 6W51, 7BH8, 7STF, 8EK5) (Stewart-Jones et al., 2009; Hwang et al., 2021; Hsiue et al., 2021; Li et al., 2022; Wright et al., 2023; Sun et al., 2023). These complexes exhibited a diversity of mutation sites, including mutations on the TCR mimic loops, the peptide, and the MHC, as well as a mix of single and multiple mutations. For each structure, we calculated the ground truth $\triangle\triangle G$ based on changes in binding affinity from the wild-type to the mutant protein. In cases where a mutation resulted in undetectable affinity by SPR, we estimated the mutant protein's affinity to be 100,000 nM—a conservative approximation given that true affinity values in such cases are often much weaker. This threshold effectively reflects a significant loss of binding, as interactions with affinities above 100,000 nM are generally considered too weak for physiological relevance (Table 4). Finally, we compared these experimental $\triangle\triangle G$ values to the predicted $\triangle\triangle G$ values generated by $\triangle\triangle G$, allowing us to assess the predictive accuracy of computational models for binding energy changes in TCR mimic systems.

*Table 4.* TCR mimic case study dataset.

| Pdb | Mutation(s) | $\Delta\Delta G$ (*kcal/mol*) | Notes |
|---|---|---|---|
| 8ek5 | EA59A | 0.428314 | HLA mutation; Figure 1J |
| 8ek5 | EA63A | No binding | HLA mutation; Figure 1J |
| 8ek5 | QA73A | 2.427024 | HLA mutation; Figure 1J |
| 8ek5 | TA74A | 1.037010 | HLA mutation; Figure 1J |
| 8ek5 | QA156A | -0.185302 | HLA mutation; Figure 1J |
| 8ek5 | TA164A | 0.020049 | HLA mutation; Figure 1J |
| 8ek5 | QC1A | 0.151418 | peptide mutation; Supp. Figure S16 |
| 8ek5 | NC3A | 1.900784 | peptide mutation; Supp. Figure S16 |
| 8ek5 | PC4A | 0.218890 | peptide mutation; Supp. Figure S16 |
| 8ek5 | IC5A | 1.853897 | peptide mutation; Supp. Figure S16 |
| 8ek5 | RC6A | No binding | peptide mutation; Supp. Figure S16 |
| 8ek5 | TC7A | 1.571367 | peptide mutation; Supp. Figure S16 |
| 8ek5 | TC8A | 0.716476 | peptide mutation; Supp. Figure S16 |
| 8ek5 | IC5L | 0.433947 | peptide mutation; estimated Kd values |
| 8ek5 | IC5V | 0.921828 | peptide mutation; estimated Kd values |
| 8ek5 | IC5G | 2.163521 | peptide mutation; estimated Kd values |
| 7stf | VC12G | No binding | peptide mutation |
| 7stf | FL53W | -0.299732 | TCRm mutation |
| 7stf | VH104N | 0.436337 | TCRm mutation |
| 7stf | VH104R | -0.421575 | TCRm mutation |
| 7stf | VH104R,VC12G | 1.431891 | TCRm and peptide mutation |
| 7bh8 | YG97S,YG98A,GG99Q,SG100Y | -1.394408 | TCRm mutations (affinity maturation) |
| 7bh8 | YG97G,YG98A,GG99Q,SG100W | -1.132139 | TCRm mutations (affinity maturation) |
| 6uj9 | QC7R | 0.760277 | peptide mutation; residue faces inside HLA groove |
| 6uj9 | YH103H,QC7R | No binding | TCRm and peptide mutation |
| 6uj9 | YH103H | 0.355047 | TCRm mutation |
| 6w51 | HF8R | No binding | peptide mutation |
| 3hae | SL26E,SL96G | -0.628096 | T1 mutant vs. 3M4E5 TCR mimic |
| 3hae | SL26E,SL96G,VC9C | -0.806352 | T1 mutant with peptide anchor residue mutation |
| 3hae | VC9C | 0.005531 | peptide anchor residue mutation only |

# B. Theoretical properties of a StaB-ddG and $\Delta\Delta G_{\text{bind}}$ predictors

In this section, we first provide a complete statement of Expressivity in Proposition 3.1. We then discuss all three properties for each of the other predictors in Table 1. Our statement of relies on a dataset of binding $\Delta\Delta G$ measurements of the form introduced in Section 3.2.

**Proposition B.1** (Proposition 3.1, Expressivity (formal))*. For any $\mathcal{D}$, there exists $\Delta b_\theta \in \mathcal{B}$ such that*

$$\Delta b_\theta(s_{n,ref}, s_{n,m}) = y_{n,m}$$

*for all* $(x_n, s_{n,ref}, s_{n,m}, y_{n,m}) \in \mathcal{D}$.

## B.1. Proof of Proposition 3.1

**Expressivity.** Consider a simplex-valued function that for each $n = 1, \ldots, N$ satisfies $p_\theta(s_{n,\text{ref}}|x_n) \propto \exp\{y_{n,\text{ref}}\}$ and for each $m = 1, \ldots, M_n, p_\theta(s_{n,m}|x_n) \propto \exp\{y_{n,m}\}$. Notice that $\log p_\theta(s_{n,\text{ref}}|x) = y_{n,\text{ref}} + c$ and $\log p_\theta(s_{n,m}|x) = y_{n,m} + c$ where $c$ is a constant. The corresponding function $\Delta b_\theta \in \mathcal{B}$ therefore satisfies $\Delta b_\theta(s_{n,\text{ref}}, s_{n,m}|x_n) = \log p_\theta(s_{n,\text{ref}}|x) - \log p_\theta(s_{n,\text{ref}}|x) = y_{n,m} - y_{n,\text{ref}}$.

## B.2. Thermodynamic properties of other predictors

**Flex ddG and FoldX.** The Flex ddG and FoldX predictors use the same thermodynamic identity in Equation 1 and Equation 2 to parametrize binding $\Delta\Delta G$, and use empirical energy functions to predict the folding $\Delta G$ terms. As such, they satisfy Antisymmetry and Mutational path independence. However, the Expressivity of the predictors is fundamentally limited by the parametric form of the empirical energy function, which cannot provide close approximations to arbitrary functions.

**RDE-Net.** RDE-Net first creates neural network embeddings $h_{wt}$ and $h_{mut}$ for the wildtype and mutant respectively. The embeddings are then used as input to another neural network, denoted by MLP. The final output is computed as $(\text{MLP}(h_{mut} - h_{wt}) - \text{MLP}(h_{wt} - h_{mut}))/2$, which enforces Antisymmetry by construction. However, since MLP is in general not a linear function, there is no guarantee on Mutational path independence. Lastly, using the same proof as above, it can be shown that the neural network parametrization satisfies Expressivity.

**PPIformer.** PPIformer uses a masked language model to model sequence likelihood. The final predictor looks similar to $\Delta f_\theta$:

$$\widehat{\Delta\Delta G} = \sum_{i \in M} \log p(\hat{c}_i = s_i \mid s_{\setminus M}) - \sum_{i \in M} \log p(\hat{c}_i = m_i \mid s_{\setminus M}).$$

In the above, $M$ is a set of mutated positions, where $s_i$ denotes the wildtype amino acid and $m_i$ denotes the mutant amino acid for position $i$. The PPIformer predictor also satisfies Antisymmetry by construction. However, it does not satisfy Mutational path independence as the conditioning information, $c_{\setminus M}$, depends on the difference between wildtype and the mutant. As such, the conditioning information between two pairs of sequences will be different. Lastly, each mutated position is predicted independently from the other mutated positions. As such, the predictor enforces the effects between any set of mutations to be additive. The enforced additivity does not satisfy Expressivity as mutations generally involve non-additive effects.

**Other predictors.** Surface-VQMAE, Prompt-DDG, DiffAffinity and ProMIM are parametrized by multilayer perceptrons that take as input embeddings from another neural network, without any guarantees of Antisymmetry or Mutational path independence. However, these predictors are expressive, treating neural networks as expressive functions.

## C. SKEMPIv2.0 filtering and metrics

In this section we outline the dataset filtering and splitting details and discuss the "per interface" metrics.

### C.1. SKEMPIv2.0 filtering and splitting procedure

The original SKEMPIv2 dataset contains 7,085 mutant entries. We filter and split SKEMPIv2.0 according to the following steps.

1. Remove 285 mutants with missing affinity measurements.

2. Remove 884 duplicate mutants with the same mutations on the same crystal structure.

3. Remove 1,029 mutants that only contain mutations at non-interface residues. We remove these because mutations at non-interface residues do not have significant contributions to binding affinity changes (Dourado & Flores, 2014).

4. Remove 108 complexes with less than 3 mutants assayed. This reduces the bias and noise from different experimental conditions.

5. Remove 4 complexes with more than 40% of the measured $\Delta\Delta G$ to be the same value. This removes 49 mutants.

6. Remove 8 complexes with unresolved residues in the crystal structure. This removes 162 mutants.

After these filtering steps, we have 201 complexes and 4,541 mutants. We cluster the complexes using the original SKEMPIv2.0 clusters based on structural homology near the binding site, resulting in 64 disjoint clusters (Jankauskaitė et al., 2019). Then, we perform a random splitting to obtain 20 clusters with 1,491 mutants across 81 complexes as our test set. We report these clusters and split at `https://github.com/LDeng0205/StaB-ddG/blob/main/data/SKEMPI/train_clusters.txt` and `https://github.com/LDeng0205/StaB-ddG/blob/main/data/SKEMPI/test_clusters.txt`.

### C.2. Additional metrics for evaluating binding prediction accuracy and cluster bootstrap intervals

We introduce additional metrics for assessing prediction accuracy: Pearson correlation, Root Mean Squared Error (RMSE), and Area Under the Receiver Characteristic (AUROC). AUROC is computed on the binary classification task of whether a mutation increases binding affinity ($\Delta\Delta G < 0$). We additionally compute the "overall" metrics for the entire set of predictions that include different complexes.

For overall metric on the SKEMPI dataset and the TCR mimic and yeast-display datasets we compute approximate standard errors as the standard deviation of that metric on cluster-bootstrap resamples of the test set where on each bootstrap sample we draw full complexes from the test set complexes with replacement (Cameron & Miller, 2015). These standard errors approximate the variability in the overall metrics owing to the choice of structures included in the test set.

The "per interface" metrics reported in Table 7 are obtained by computing each metric for each complex, then take the average across complexes. For complexes that contain less than 10 mutants, the correlation values obtained are empirically observed to be noisy (Figure 5). As such, we decide to report the mean of metrics for complexes with 10 or more mutants to reduce the effect of noise. We examine the impact of the choice of such a threshold on the relative performance between STAB-DDG andm Flex ddG (Figure 6). We find that the relative performance is robust to the choice of the threshold.

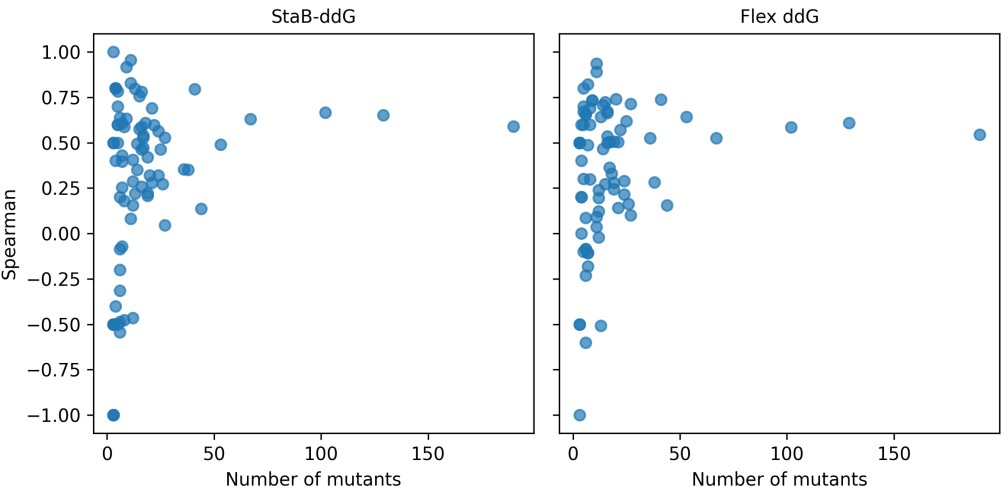

*Figure 5.* Spearman correlation vs. Number of mutants. Each point represents a complex.

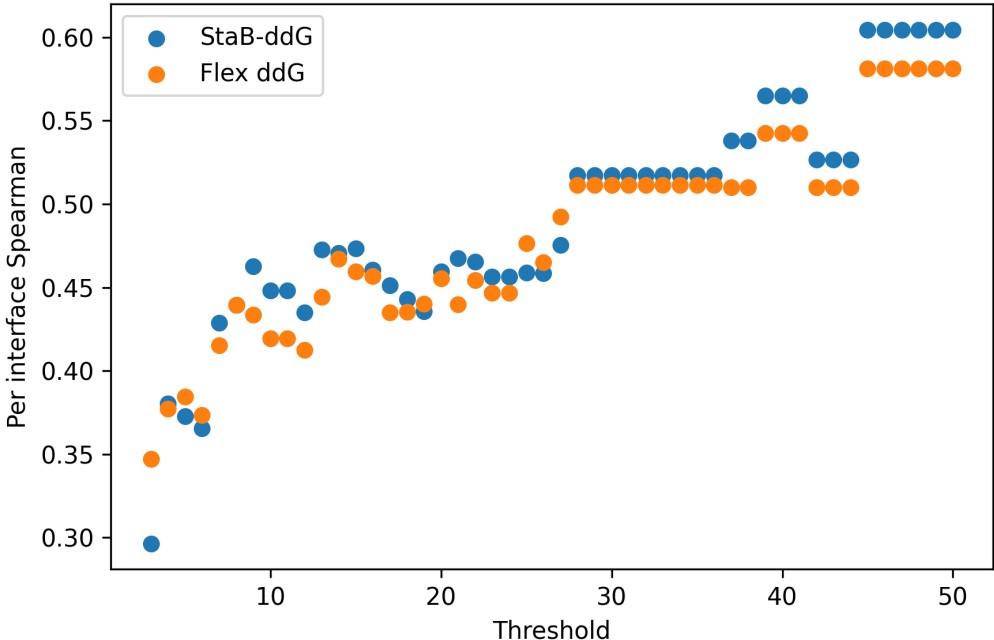

*Figure 6.* The relative performance of models for different thresholds when computing per interface spearman.

## D. Additional experiments on variations of STAB-DDG

### D.1. Linear model initialization

Tsuboyama et al. (2023) proposed to use amino acid-specific offsets in a model relating sequence probabilities in protein families to stability measurements. We experimented with a similar approach and introduce a linear model on top of our folding $\Delta\Delta G$ predictor as

$$F(s, s') = \alpha \Delta f_\theta(s, s') + (\sum_{a \in s} \phi_a - \sum_{a \in s'} \phi_a) + \phi_0 \tag{9}$$

where $\alpha$ is a scaling term to correct for the initial scale mismatch, $\phi_0$ is a global bias, and $\phi_a$ represents amino-acid specific offsets. We fit the linear parameters first with the zero-shot predictor $\Delta f_\theta(s, s')$ before fine-tuning $\theta$. This procedure is inspired by the idea that fine-tuning the last layers of a neural network first could help improve generalization performance (Kumar et al., 2022). We use a predictor of the following form

$$B(s, s') = \alpha \Delta b_\theta(s, s') + (\sum_{a \in s} \phi_a - \sum_{a \in s'} \phi_a) + \phi_0 \tag{10}$$

which uses the same set of linear weights as $F$. Note that the linear model introduces asymmetry and the updated predictors no longer satisfy the first two properties of Proposition 3.1.

We found that fitting the linear model and following the same fine-tuning procedure, with $F$ and $B$ as folding and binding predictors, did not lead to a significant difference in prediction accuracy (Table 6).

We provide the fitted linear parameters $\alpha$, $\phi_0$, and $\phi_a$ in Table 5. The learned $\phi_0$ is negative, indicating that most mutations in the dataset are destabilizing. In addition, the offset for TRP, a bulky hydrophobic residue expected to have larger effects on stability, is the second largest in magnitude and is negative (destabilizing).

*Table 5.* Linear parameter values.

| $\alpha$ | $\phi_0$ | ALA | CYS | ASP | GLU | PHE | GLY | HIS | ILE | LYS | LEU | MET |
|---|---|---|---|---|---|---|---|---|---|---|---|---|
| 0.24 | -0.19 | 0.00 | -0.80 | 0.04 | 0.10 | -0.55 | 0.13 | -0.20 | -0.40 | 0.12 | -0.33 | -0.47 |

| | ASN | PRO | GLN | ARG | SER | THR | VAL | TRP | TYR |
|---|---|---|---|---|---|---|---|---|---|
| | 0.01 | 0.33 | -0.01 | -0.23 | 0.00 | -0.07 | -0.27 | -0.68 | -0.50 |

### D.2. Using AlphaFold 3 predictions for apo structures

We experimented with using AlphaFold 3 (Abramson et al., 2024) predicted structures for individual binders instead of obtaining them from the complex crystal structure. We hypothesized that this would more closely track the apo (unbound) state of the structures for more accurate folding energy predictions. We found this to have not made a significant difference in the Per interface Spearman metric, but have improved several other metrics (Table 6).

### D.3. Normalizing complex loss with the number of mutants

In our fine-tuning objective (Equation (8)), we weight each complex $n$ by the number of mutants assayed $M_n$. We additionally experimented with weighting the loss by $\sqrt{M_n}$ instead of $M_n$. However, we did not find a significant difference between the two weighting schemes (Table 6).

### D.4. Variance reduction

We evaluated the effects of antithetic variates and the number of Monte Carlo samples on reducing prediction error, measured by RMSE (Figure 7). We found that the antithetic variates method significantly reduced prediction error, and Monte Carlo ensembling further reduced the error. In addition to improving prediction accuracy at inference time, fixing the decoding order and backbone noise also led to better training dynamics. In particular, under the same hyperparameters, a model trained without fixing these additional parameters performed much worse than STAB-DDG (Table 6). In our other experiments, we ensemble over 20 samples.

## D.5. Ablations

**Fine-tuning on folding stability data without inverse-folding pre-training.**   We randomly initialized weights to our predictor and fine-tuned it on folding stability data. We found that the performance was much worse than STAB-DDG zero-shot, suggesting that pre-training contributes significantly to model performance (Table 6).

**Fine-tuning on binding affinity data from ProteinMPNN weights.**   We directly fine-tuned STAB-DDG zero-shot on binding energy data without incorporating folding stability data. We found that though the per interface metrics remained the same, the overall accuracy dropped (Table 6).

*Table 6.* Evaluation of prediction performance of variations of STAB-DDG on the test split of SKEMPIv2.0. Per interface metrics for which the difference from STAB-DDG is not statistically significant are underlined. Statistical significance is determined by a paired, one-sided t-test against STAB-DDG. Standard errors are also reported for per interface metrics.

| Method | Per Interface | | | Overall | | | |
| | Pearson | Spearman | RMSE | Pearson | Spearman | RMSE | AUROC |
|---|---|---|---|---|---|---|---|
| StaB-ddG | $0.49 \pm 0.04$ | $0.45 \pm 0.04$ | $1.41 \pm 0.12$ | $0.53 \pm 0.06$ | $0.53 \pm 0.06$ | $1.72 \pm 0.11$ | $0.73 \pm 0.05$ |
| No folding | $\underline{0.50 \pm 0.04}$ | $\underline{0.46 \pm 0.04}$ | $\underline{1.53 \pm 0.13}$ | $0.47 \pm 0.07$ | $0.47 \pm 0.05$ | $1.92 \pm 0.13$ | $0.70 \pm 0.04$ |
| No pre-train | $0.12 \pm 0.05$ | $0.12 \pm 0.05$ | $2.04 \pm 0.15$ | $0.23 \pm 0.07$ | $0.17 \pm 0.05$ | $2.41 \pm 0.14$ | $0.61 \pm 0.04$ |
| No ant. var. | $0.30 \pm 0.04$ | $0.30 \pm 0.04$ | $1.96 \pm 0.16$ | $0.19 \pm 0.08$ | $0.17 \pm 0.06$ | $2.35 \pm 0.17$ | $0.58 \pm 0.05$ |
| Linear model | $\underline{0.47 \pm 0.04}$ | $0.44 \pm 0.04$ | $\underline{1.52 \pm 0.14}$ | $0.54 \pm 0.05$ | $0.49 \pm 0.05$ | $1.79 \pm 0.11$ | $0.72 \pm 0.04$ |
| Pred. apo structures | $\underline{0.45 \pm 0.05}$ | $\underline{0.43 \pm 0.04}$ | $\underline{1.43 \pm 0.10}$ | $0.60 \pm 0.05$ | $0.56 \pm 0.05$ | $1.66 \pm 0.09$ | $0.76 \pm 0.03$ |
| sqrt(M) weighting | $\underline{0.49 \pm 0.04}$ | $\underline{0.45 \pm 0.04}$ | $\underline{1.40 \pm 0.12}$ | $0.54 \pm 0.06$ | $0.54 \pm 0.06$ | $1.72 \pm 0.12$ | $0.74 \pm 0.05$ |

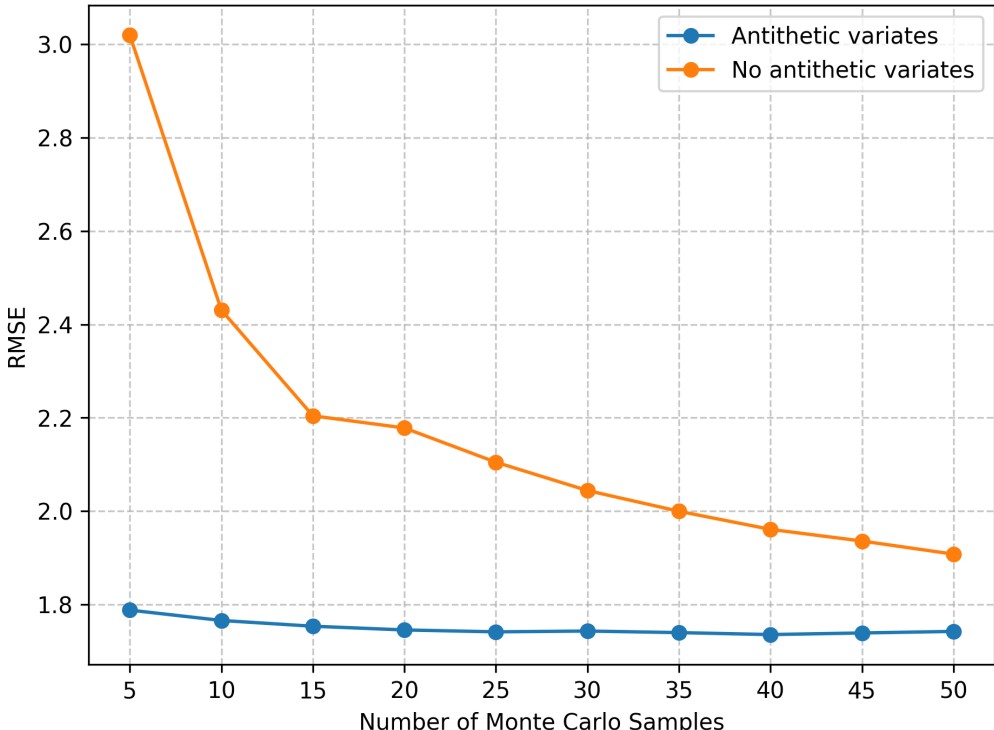

*Figure 7.* Overall RMSE vs. number of Monte Carlo samples evaluated using STAB-DDG parameters on the SKEMPIv2.0 test split.

# E. Additional details on the main text results

## E.1. Binding prediction accuracy on SKEMPIv2.0

### E.1.1. COMPARISON TO BASELINE METHODS ON MORE METRICS

Here, we provide a more complete set of metrics to compare with other methods (Table 7). In addition, we include the result for the averaged predictions of STAB-DDG and FoldX (StaB-ddG + FoldX).

### E.1.2. RMSE BY AMINO ACID TYPE

We provide the average RMSE by mutant amino acid type in Table 8. We found that STAB-DDG achieved lower RMSEs for many bulky hydrophobic residues (PHE, TRP, TYR, MET).

*Table 7.* Evaluation of baseline $\Delta\Delta G$ prediction methods and STAB-DDG on the test split of the SKEMPIv2.0 dataset. With the exception of the ensemble model (StaB-ddG + FoldX), the best approach for each metric is in **bold** and per interface metrics for which the difference from the best is not statistically significant ($P < 0.05$) are underlined. Statistical significance is determined by a paired, one-sided t-test to the best performing method. Standard errors for overall metrics are computed through cluster-bootstrapping.

| Method | Per Interface | | | Overall | | | |
|---|---|---|---|---|---|---|---|
| | Pearson | Spearman | RMSE | Pearson | Spearman | RMSE | AUROC |
| RDE-Net | $0.30 \pm 0.05$ | $0.28 \pm 0.05$ | $1.53 \pm 0.12$ | $0.40 \pm 0.05$ | $0.40 \pm 0.05$ | $1.81 \pm 0.10$ | $0.63 \pm 0.03$ |
| Surface-VQMAE | $0.35 \pm 0.05$ | $0.33 \pm 0.05$ | $1.48 \pm 0.11$ | $0.45 \pm 0.04$ | $0.44 \pm 0.05$ | $1.76 \pm 0.09$ | $0.65 \pm 0.03$ |
| ProMIM | $0.19 \pm 0.06$ | $0.15 \pm 0.05$ | $1.57 \pm 0.12$ | $0.35 \pm 0.06$ | $0.35 \pm 0.06$ | $1.85 \pm 0.11$ | $0.60 \pm 0.03$ |
| Prompt-DDG | $0.32 \pm 0.04$ | $0.27 \pm 0.04$ | **$1.41 \pm 0.12$** | $0.33 \pm 0.08$ | $0.35 \pm 0.07$ | $1.81 \pm 0.13$ | $0.57 \pm 0.05$ |
| DiffAffinity | $0.26 \pm 0.04$ | $0.25 \pm 0.04$ | $1.55 \pm 0.13$ | $0.31 \pm 0.05$ | $0.33 \pm 0.05$ | $1.88 \pm 0.11$ | $0.64 \pm 0.04$ |
| PPIformer | $0.20 \pm 0.04$ | $0.20 \pm 0.04$ | $1.51 \pm 0.10$ | $0.46 \pm 0.07$ | $0.42 \pm 0.06$ | $1.77 \pm 0.09$ | $0.71 \pm 0.05$ |
| ProteinMPNN | $0.14 \pm 0.04$ | $0.13 \pm 0.04$ | — | $0.18 \pm 0.07$ | $0.18 \pm 0.06$ | — | $0.55 \pm 0.04$ |
| ESM-IF1 | $0.24 \pm 0.04$ | $0.22 \pm 0.04$ | — | $0.15 \pm 0.05$ | $0.23 \pm 0.08$ | — | $0.54 \pm 0.05$ |
| Flex ddG | $0.45 \pm 0.04$ | $0.42 \pm 0.04$ | $1.93 \pm 0.50$ | $0.22 \pm 0.17$ | $0.54 \pm 0.05$ | $3.98 \pm 1.65$ | $0.74 \pm 0.03$ |
| FoldX | **$0.49 \pm 0.03$** | **$0.48 \pm 0.03$** | $1.63 \pm 0.12$ | **$0.54 \pm 0.05$** | **$0.56 \pm 0.05$** | $1.92 \pm 0.12$ | **$0.77 \pm 0.04$** |
| StaB-ddG zero-shot | $0.45 \pm 0.04$ | $0.43 \pm 0.04$ | — | $0.44 \pm 0.07$ | $0.43 \pm 0.06$ | — | $0.68 \pm 0.04$ |
| Stability fine-tuned | $0.45 \pm 0.04$ | $0.40 \pm 0.04$ | $1.69 \pm 0.15$ | $0.44 \pm 0.06$ | $0.45 \pm 0.06$ | $2.00 \pm 0.12$ | $0.70 \pm 0.04$ |
| StaB-ddG | **$0.49 \pm 0.04$** | $0.45 \pm 0.04$ | **$1.41 \pm 0.12$** | $0.53 \pm 0.06$ | $0.53 \pm 0.05$ | **$1.72 \pm 0.11$** | $0.73 \pm 0.04$ |
| StaB-ddG + FoldX | $0.56 \pm 0.04$ | $0.53 \pm 0.03$ | $1.32 \pm 0.12$ | $0.59 \pm 0.05$ | $0.61 \pm 0.05$ | $1.62 \pm 0.11$ | $0.78 \pm 0.04$ |

*Table 8.* RMSE for each mutant amino acid.

| ALA | CYS | ASP | GLU | PHE | GLY | HIS | ILE | LYS | LEU | MET |
|---|---|---|---|---|---|---|---|---|---|---|
| 1.32 | 3.19 | 2.96 | 1.96 | 0.94 | 1.33 | 2.06 | 1.90 | 1.61 | 1.67 | 1.58 |

| ASN | PRO | GLN | ARG | SER | THR | VAL | TRP | TYR |
|---|---|---|---|---|---|---|---|---|
| 1.37 | 0.99 | 1.93 | 1.78 | 0.81 | 1.04 | 1.14 | 0.82 | 1.44 |

## E.2. Yeast surface display case study details

**Estimation of a proxy for binding $\Delta\Delta G$ from sequence counts.** We briefly summarize the procedure of estimating $\Delta\Delta G$ from yeast surface display sorts described fully in Cao et al. (2022). A midpoint concentration ($SC_{50}$) is estimated as a proxy for the binding dissociation constant $K_D$ used to compute $\Delta G_{\text{bind}}$. The $SC_{50,i}$ for sequence $i$ is estimated using (Equation (1) Cao et al. (2022))

$$\text{Fraction\_collected}_i = \frac{\text{concentration}}{(\text{concentration} + SC_{50,i})}.$$

Here, Fraction_collected$_i$ is the fraction of bound sequences as determined by Fluorescence-Activated Cell Sorting (FACS) and Next Generation Sequencing (NGS). A critical assumption in this procedure is that expression level is constant across different sequences.

**Binding confounded by expression.** We found that our folding stability predictor was more accurate than our binding energy predictor at predicting binding energy on the yeast surface display dataset (Figure 4). We reasoned that this effect could be attributed to the sequence count readout from the yeast surface display experiment depended on both binding affinity and expression, a quantity strongly correlated with folding stability (Cao et al., 2022). We validated this hypothesis by experimenting with predictors of the form

$$b_\theta(s_{A:B}) = f_\theta(s_{A:B}) - \beta[f_\theta(s_A) + f_\theta(s_B)].$$

The case that $\beta = 0$ corresponds to the "complex" only predictions. And the case that $\beta = 1$ corresponds to STAB-DDG parameterization. where $\beta$ is a constant. We found that, indeed, setting $\beta = 0.65$ improved performance of all predictors (Figure 8).

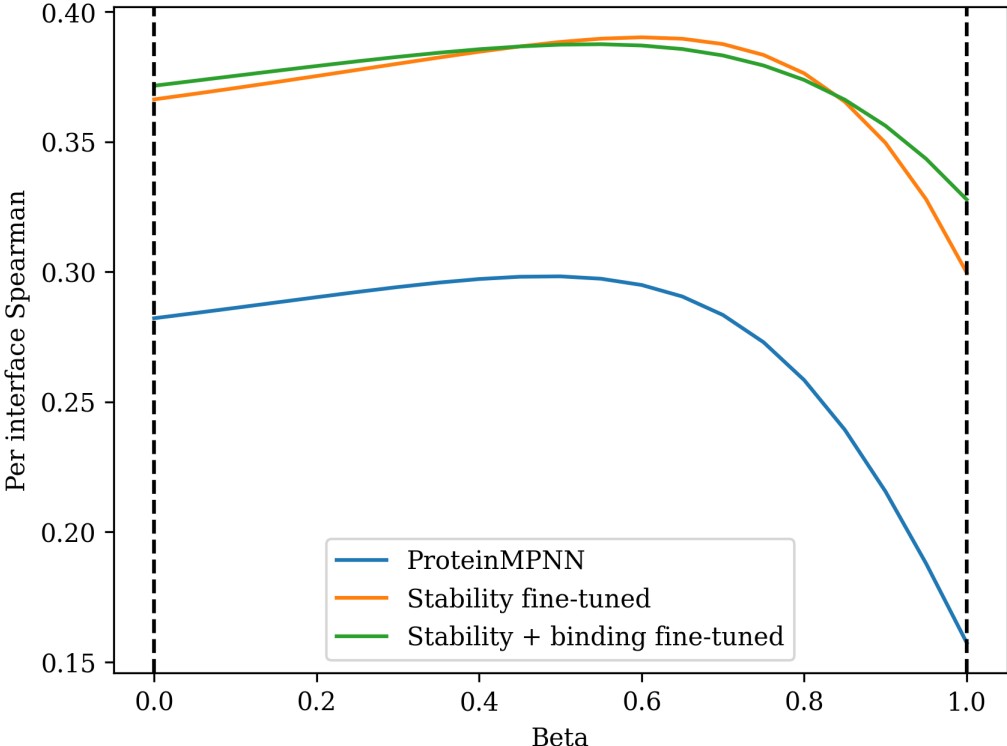

*Figure 8.* Spearman vs. different values of $\beta$ on the yeast surface display dataset. $\beta = 0$ corresponds to the folding energy predictor $\Delta f_\theta$, and $\beta = 1$ corresponds to the binding energy predictor $\Delta b_\theta$.

*Table 9.* Evaluation of $\Delta\Delta G$ prediction on yeast surface display dataset.

| Method | Per Interface | | | Overall | | | |
|---|---|---|---|---|---|---|---|
| | Pearson | Spearman | RMSE | Pearson | Spearman | RMSE | AUROC |
| StaB-ddG zero-shot | $0.157 \pm 0.020$ | $0.094 \pm 0.015$ | $1.43 \pm 0.08$ | 0.16 | 0.10 | 1.45 | 0.54 |
| StaB-ddG zero-shot complex | $0.282 \pm 0.023$ | $0.274 \pm 0.025$ | $3.02 \pm 0.09$ | 0.25 | 0.26 | 3.08 | 0.61 |
| Stability fine-tuned | $0.300 \pm 0.022$ | $0.268 \pm 0.021$ | $1.24 \pm 0.10$ | 0.28 | 0.26 | 1.30 | 0.61 |
| Stability fine-tuned complex | $0.366 \pm 0.030$ | $0.314 \pm 0.030$ | $1.26 \pm 0.07$ | 0.31 | 0.29 | 1.26 | 0.62 |
| StaB-ddG | $0.328 \pm 0.025$ | $0.296 \pm 0.026$ | $1.21 \pm 0.09$ | 0.32 | 0.28 | 1.25 | 0.62 |
| StaB-ddG complex | $0.372 \pm 0.030$ | $0.323 \pm 0.031$ | $1.35 \pm 0.06$ | 0.32 | 0.30 | 1.35 | 0.63 |

## E.3. Relative performance of DSMbind

Jin et al. (2023) have previously demonstrated strong zero-shot $\Delta\Delta G_{\text{bind}}$ prediction on the SKEMPI dataset. They report an overall (rather than average per-interface) Spearman correlation of 0.42 (Jin, 2025). This result is comparable to the best

performing supervised deep-learning baselines Surface-VQMAE and PPIformer, but is significantly below that of FoldX and Stab-ddG (Table 7). While Jin et al. (2023) remark that DSMBind performs comparably to FoldX, we suspect this discrepancy with our results owes to a difference in running settings of FoldX, in particular the number of FoldX `repair` step before prediction. We were unsuccessful in an attempt to make a more direct comparison to DSMbind; multiple attempts to run the public code (we tried different versions of dependencies for which the required releases were not specified) led to crashes or `NaN` predictions.

### E.4. Comparison to ThermoMPNN

In this section we compare the performance of STAB-DDG on folding stability prediction with ThermoMPNN, a state-of-the-art method for predicting the effects of single mutations on protein stability. ThermoMPNN is based on ProteinMPNN and introduces an additional attention-based neural network for fine-tuning on the Megascale dataset. We use the same training split as ThermoMPNN to fine-tune STAB-DDG. However, as STAB-DDG naturally generalizes to mutants with multiple mutations, we include such mutants for structures in the training set. We evaluate STAB-DDG on the same test set as ThermoMPNN (Table 10). We report the metrics computed on the test set as a whole, rather than averaging performance across domains. In addition to ThermoMPNN, we include the next best two baselines reported by Dieckhaus et al. (2024) for reference. STAB-DDG achieves performance not much below that of ThermoMPNN and outperforms the next best baseline (Dieckhaus et al., 2024). We also assess the performance on multiple mutations for our method (Table 11). We found that though our method performed comparably to ThermoMPNN on single mutations, our stability predictor achieved significantly lower accuracy on multiple mutations, presumably due to the limited amount of multiple mutation data.

*Table 10.* Performance on Megascale test set (single mutations).

| METHOD | PEARSON | SPEARMAN | RMSE |
|---|---|---|---|
| THERMOMPNN | 0.75 | 0.73 | 0.71 |
| STABILITY FINE-TUNED (OURS) | 0.71 | 0.69 | 0.77 |
| RASP | 0.71 | 0.67 | 1.08 |
| PROSTATA | 0.64 | 0.59 | 0.83 |

*Table 11.* Performance on Megascale test set (multiple mutations).

| METHOD | PEARSON | SPEARMAN | RMSE |
|---|---|---|---|
| STABILITY FINE-TUNED (OURS) | 0.38 | 0.42 | 1.41 |

