# OpenReview forum: "Predicting mutational effects on protein binding from folding energy"
_ICML.cc/2025/Conference — ICML 2025 poster_

### Official Review · Reviewer_J9eu · 2025-03-10

**Overall Recommendation:** 4

**Summary:**

The authors propose a deep learning method for predicting binding energy differences (i.e. ΔΔGs) for closely related pairs of protein-protein complexes. To do so the authors rely on a well-known identity that relates binding energy differences to folding free energies. The authors then proceed to fine-tune a pre-trained inverse folding model--a proxy for a folding energy predictor--on both folding and binding energy data. In their empirical evaluation the authors demonstrate that the resulting predictor, STAB-DDG, is competitive with, and may indeed outperform, the state-of-the-art Rosetta-based predictor Flex ddG.

**Claims And Evidence:**

Yes the claim that STAB-DDG provides reasonably good ΔΔG predictions (this remains a difficult problem, despite the author's contributions) is well supported by the empirical evaluation. In particular the authors seem to have taken care to minimize possible issues with data leakage, and have taken the time to curate an additional test set for consideration (TCR mimics). (Though I should note that I have not examined the information in the supplementary materials about the data splitting strategy).

Granted, the limited size of test sets limits the ability to do fine-grained evaluation; nevertheless, it would be great if the authors made some attempt to do so. For example, does STAB-DDG tend to do worse on larger complexes? On certain classes of protein? Aggregate metrics are great, but it would be great if the authors could expand their results to give a better understanding of their method's common failure modes.

**Essential References Not Discussed:**

N/A

**Experimental Designs Or Analyses:**

As far as I can tell the experimental designs/analyses look sound.

**Methods And Evaluation Criteria:**

The experimental method and reported methods appear to follow best practices in the field. As stated above, it would be great if the authors could offer more nuanced results that go beyond aggregate metrics.

**Other Comments Or Suggestions:**

- In Eqn. 5 $M_n$ can presumably be quite variable in practice, with the consequence that some data points will contribute much less to the loss than others. Have you considered using a loss that doesn't normalize by $M_n$? For example, one could normalize by $\sqrt{M_n}$.
- You state that "We fit the linear parameters first with the zero-shot predictor Δfθ(s, s′) before fine-tuning θ.". Can you please explain your rationale for doing so in more detail? Why not fit jointly? Why do you prefer a two-stage approach? Have you tried both approaches?
- What do you think is the driving factor explaining the (smallish) performance differences between your results and those of Dieckhaus 2024? Is it their particular architecture? Is it something else?
- You state that "Importantly, the imbalance in the numbers of mutants across different structures introduces bias to the “Per-Structure” metrics". I guess this would more accurately be described as variance?

typos:
- double period in prop 3.1 last bullet point
- line 200: "denote the a"

**Other Strengths And Weaknesses:**

The paper is generally well-written and easy to follow. While there is nothing particularly surprising about the approach taken, the authors do a good job of motivating the rationale behind their approach and demonstrating that it can perform well in practice. As such I think submission could be of interest to the ICML community and recommend acceptance.

**Questions For Authors:**

N/A

**Relation To Broader Scientific Literature:**

The discussion of related work in Sec 4 is pretty thorough, including a discussion of both classical and ML-based approaches.

**Theoretical Claims:**

The (straightforward) theoretical results included in the submission, specifically Proposition 3.1, look sound.

In this context I note that the authors comment that "the linear model introduces asymmetry". As I understand this is entirely driven by $\phi_0 \ne 0$. Can you please comment and extend your discussion on this point? What value of $\phi_0$ do you learn in practice? Does keeping this term actually make any appreciable difference to the empirical performance?

---

> ### Author Rebuttal · Authors · 2025-04-01
>
> Thank you for your review. We appreciate your thoughtful suggestions and have conducted additional experiments that provide additional insights to our method.
>
> Fine-grained evaluation: We examine StaB-ddG’s performance on different subsets of SKEMPI.
>
> Complex size: total number of residues of a PPI.
>
> (PS RMSE: Per Structure RMSE)
>
> | Complex Size        | PS RMSE |
> |---------------------|---------|
> | < 200 residues      | 1.05    |
> | < 400 residues      | 1.33    |
> | < 600 residues      | 1.48    |
> | < 800 residues      | 1.42    |
> | < 1000 residues     | 1.51    |
>
> Interface structural rigidity: percent of near-interface residues (defined as within 10A of another chain) with secondary structure type = loop.
>
> | Interface Rigidity | PS RMSE |
> |--------------------|---------|
> | < 30% loops        | 1.254   |
> | < 40% loops        | 1.414   |
> | < 50% loops        | 1.465   |
> | < 60% loops        | 1.508   |
> | < 70% loops        | 1.492   |
> | < 80% loops        | 1.496   |
>
> We observe that StaB-ddG does worse on bigger complexes and more flexible interfaces. We believe that this information provides valuable additional insight to our method and will include it in the appendix.
>
> Learned linear parameters (negative means destabilizing):
>
> $\alpha = 0.24, \phi_0 = -0.19$
>
> Amino acid offsets
>
> | ALA | CYS | ASP | GLU | PHE | GLY | HIS | ILE | LYS | LEU | MET | ASN | PRO | GLN | ARG | SER | THR | VAL | TRP | TYR |
> |-----|-----|-----|-----|-----|-----|-----|-----|-----|-----|-----|-----|-----|-----|-----|-----|-----|-----|-----|-----|
> | 0.00 | -0.80 | 0.04 | 0.10 | -0.55 | 0.13 | -0.20 | -0.40 | 0.12 | -0.33 | -0.47 | 0.01 | 0.33 | -0.01 | -0.23 | 0.00 | -0.07 | -0.27 | -0.68 | -0.50 |
>
> Since both the fine-tuning losses and ddG predictions are invariant to additive shifts that might be applied to all per-amino-acid offsets, we report the differences relative to Alanine. We see that the learned $\phi_0$ is negative, indicating that most mutations in the dataset are destabilizing. We note that the value of the bias does not affect the correlation metrics, and find that adding the bias term lowers the overall RMSE from 1.88 to 1.79. We also note that the learned offset for TRP, a bulky hydrophobic residue expected to have larger effects on stability, is the second largest in magnitude and is negative (destabilizing).
>
> Two-stage finetuning: Our rationale for our two-stage finetuning is that first fitting the linear parameters can correct for any initial scale mismatch in our predictions that might inflate initial gradients and destabilize training.  When we view the linear model as the final layer of a deep predictor, this choice coincides with standard practice in transfer learning [1]. In our revision we will elaborate on this choice.
>
> Comparison to ThermoMPNN: We believe the driving factor of the performance difference between our model and ThermoMPNN is (1) the architectural difference, and (2) we train on multiple mutations as well while ThermoMPNN are trained only on single mutations.
>
> $\sqrt{M_n}$ : Thank you for your suggestion to try our loss with $\sqrt{M_n}$ . We re-trained the model with this loss and did not observe a meaningful difference with our original loss, we will include the results below in the appendix:
> |                     | $M_n$    | $\sqrt{M_n}$   |
> |------------------|-------|-------|
> | Overall Pearson  | 0.553 | 0.548 |
> | Overall Spearman | 0.515 | 0.515 |
>
> Bias and variance in per-structure metrics: For structures with a small number of variants computed correlations are subject to both significant bias and high variance (see e.g. [2]).  Though non-exhaustive, we feel our figure 5 in our submission gives some indication of these effects as we have no reason to expect performance to be different on structures with few mutants.
>
> Typos: Thanks for pointing these out! We will fix them accordingly in the revision.
>
> [1] Kumar et al. Fine-tuning can Distort Pre-trained Features and Underperform Out-of-Distribution. ICLR 2022.
>
> [2] Bishara, Anthony J. and Hittner, James B. Reducing Bias and Error in the Correlation Coefficient Due to Nonnormality. Educational and Psychological Measurement.

---

### Official Review · Reviewer_aZe7 · 2025-03-12

**Overall Recommendation:** 4

**Summary:**

This work introduces Stab-DDG, a deep-learning method for DDG prediction that leverages both folding energy and binding energy data during pre-training. The authors relate folding energy to binding energy in a principled manner, which leads to a loss function for pre-training on folding energy data specifically. The authors show that Stab-DDG and its linear variant outperform existing methods on the SKEMPI v2.0 dataset and a TCR mimic case study. The authors also contribute theoretical criteria for DDG predictors overall and show that their proposed method fits these criteria well.

**Claims And Evidence:**

The claims of the paper are generally well-supported by the experiments and theoretical analysis. The authors show that Stab-DDG outperforms existing methods on the SKEMPI v2.0 dataset and a TCR mimic case study. The authors also show that Stab-DDG fits the theoretical criteria for DDG predictors well, and the connection between folding and binding energy is fairly straightforward.

**Essential References Not Discussed:**

Besides the missing baselines in the experimental section, I think all relevant references are already discussed.

**Experimental Designs Or Analyses:**

**Strengths**
- The overall experimental designs are sound.
- The authors correctly point out issues in dataset splitting in SKEMPI v2.0 from previous works and provide a solution with documented filtering steps.
- The authors include standard errors in their tables with statistical significance tests for per-structure Pearson/Spearman coefficients, which is good practice.

**Weaknesses**
- It's not clear why similar significance testing is not done for the overall metrics in Table 4.
- There is no mention of specific hyperparameters such as hidden state sizes, epoch numbers, learning rates, initializations, optimizers, etc. for each step of the pre-training and fine-tuning procedures, which are crucial for reproducibility.
- Similar to the Methods section, there are no parameter sensitivity studies for $\alpha$ and the number of noise terms $\epsilon$.
- The authors include comparisons with RDE-PPI and PPIFormer, which are strong baselines. However, there is a wealth of other deep learning-based methods like DiffAffinity [1], Prompt-DDG [2], Surface-VQMAE [3], Boltzmann Alignment [4], ProMIM [5] etc. that are not compared against. I think this is especially important since the authors are offering not only a new method but also a new split and filtering method for SKEMPI v2.0, and it would be helpful to see how Stab-DDG compares to these other methods under this new setting.

[1] Liu et al. Predicting mutational effects on protein-protein binding via a side-chain diffusion probabilistic model. NeurIPS 2023.

[2] Wu et al. Learning to Predict Mutational Effects of Protein-Protein Interactions by Microenvironment-aware Hierarchical Prompt Learning. ICML 2024.

[3] Wu, Fang and Li, Stan Z. Surface-VQMAE: Vector-quantized Masked Auto-encoders on Molecular Surfaces. ICML 2024.

[4] Jiao et al. Boltzmann-Aligned Inverse Folding Model as a Predictor of Mutational Effects on Protein-Protein Interactions. ICLR 2025.

[5] Mo et al. Multi-level Interaction Modeling for Protein Mutational Effect Prediction. Preprint.

**Methods And Evaluation Criteria:**

**Strengths**
- The proposed model and evaluation metrics for SKEMPI v2.0 and the TCR mimic case are all sensible and customary for the DDG prediction task.
- Leveraging folding energy data for pre-training is a novel and interesting approach that is well-motivated.

**Weaknesses**
 - The authors mention that a scalar $\alpha$ is used to scale the inverse folding model's output, and that averaging over noise terms during inference is used to reduce variance, but the exact values of $\alpha$ and the number of noise terms $\epsilon$ are not provided. Is $\alpha$ learned?

- If I'm understanding correctly, Stab-DDG requires estimating log-probabilities of both bound complexes and unbound monomers. However, to my knowledge, SKEMPI v2.0 only contains data for bound complexes. It would be helpful to clarify how unbonded monomer data is obtained or if it is used at all for SKEMPI v2.0.

**Other Comments Or Suggestions:**

I think this work is well-motivated and a valuable contribution. However, I think the main weaknesses are more about completeness in the experimental section, e.g. lack of hyperparameter and optimization details and missing deep learning baselines.

**Other Strengths And Weaknesses:**

**Weaknesses**

- To my knowledge, the core method and architecture, using the log odds-ratios of wild-type and mutant sequences, is not novel. Other works like Boltzmann Alignment [1] and ProteinMPNN-DDG [2] have used similar approaches using the same backbone inverse folding model to predict DDG.

[1] Jiao et al. Boltzmann-aligned inverse folding model as a predictor of mutational effects on protein-protein interactions. ICLR 2025.

[2] Dutton et al. Improving Inverse Folding models at Protein Stability Prediction without additional Training or Data. MLSB @ NeurIPS 2024.

**Questions For Authors:**

- Do the authors observe any interesting per-residue patterns in model predictions on the SKEMPI v2.0 dataset? For example, are there certain residues that the model consistently underestimates or overestimates the $\Delta \Delta G$ for?
- While Stab-DDG is used for DDG prediction, what kind of performance does it achieve on just "DG", or binding affinity prediction on datasets like SAbDab?

**Relation To Broader Scientific Literature:**

To my knowledge, this is the first work to directly connect folding energy to binding energy for deep learning-based DDG prediction.

**Theoretical Claims:**

All proofs referenced in Appendix B are straightforward to check and are sound.

---

> ### Author Rebuttal · Authors · 2025-04-01
>
> Thank you for your review. We appreciate your recognition that our use of folding energy data to improve binding prediction is novel and well-motivated. We found your suggestions helpful and address them below.
>
> Baselines: We agree with the reviewer’s comment that the lack of deep learning baselines was a limitation of our submission.  This concern was shared by reviewer yFSX, and we address it by including four new baselines. The results show that on our stricter split these additional methods provide similarly poor generalization performance. Please see our reply to yFSX for details.
>
> Reproducibility and hyperparameters: Thank you for pointing out these missing details, which will be included in the revision.  In brief, the network architecture and initialization are inherited exactly from ProteinMPNN (1.6M parameters, 3 layers for both encoder and decoder with hidden state size = 128) [1].  We fine-tune on the megascale stability dataset using the ADAM optimizer with learning rate 3e-5 for 150 epochs with a batch size of 50,000 tokens. We fine-tune on SKEMPI using the ADAM optimizer with learning 1e-6 for 50 epochs with a batch size of 50,000 tokens. To further ensure reproducibility our final version will include training and inference code.
>
> Value of alpha: alpha is a scaling term learned from folding stability data and we will include them in the appendix (negative means destabilizing)
>
> $\alpha = 0.24, \phi_0 = -0.19$
>
> We see that the learned bias ($\phi_0$) is negative, indicating that most mutations in the dataset are destabilizing.
>
> Per-residue patterns: we report the RMSE by mutant residue type for single mutants on SKEMPI below.
> | ALA | CYS | ASP | GLU | PHE | GLY | HIS | ILE | LYS | LEU | MET | ASN | PRO | GLN | ARG | SER | THR | VAL | TRP | TYR |
> |-----|-----|-----|-----|-----|-----|-----|-----|-----|-----|-----|-----|-----|-----|-----|-----|-----|-----|-----|-----|
> | 1.33 | 3.54 | 2.96 | 1.89 | 0.99 | 1.52 | 1.37 | 1.83 | 1.52 | 1.92 | 1.45 | 1.45 | 1.09 | 2.07 | 1.73 | 1.01 | 1.20 | 0.96 | 1.00 | 1.22 |
>
> We note that StaB-ddG achieves lower RMSEs for many bulky hydrophobic residues (F, W, Y, M).
>
> Size of ensemble over noise: we performed an ablation experiment and will include a figure for the impact of the size of ensemble in the appendix. We report it partially here:
>
> | Ensemble Size | Overall Pearson |
> |---------------|-----------------|
> | 1             | 0.524           |
> | 3             | 0.528           |
> | 5             | 0.536           |
> | 10            | 0.546           |
> | 15            | 0.549           |
> | 20            | 0.544           |
> | 40            | 0.548           |
>
> We find that ensembling over 10 predictions with random permutation orders and backbone noise leads to near optimal performance. The results in table 4 are obtained with ensemble size = 20.
>
> Monomers for Skempi: are obtained by splitting the PDB files by chain (holo structures). We will update the text to clarify this.
>
> Standard errors and significance testing on per-structure and overall metrics: we’re glad that the reviewer appreciates that we reported standard errors on the per-structure metrics.  For the revision we will include cluster-bootstrap standard errors for overall metrics, and (because their interpretation is non-trivial) a discussion of them in the appendix.
>
> In brief, we initially chose not to report intervals for overall metrics because it was not clear what the standard error should represent.  For each per-structure metric, the standard error represents uncertainty in the expected value of that metric for a collection of ddG measurements for mutants of some new structure typical of those in the test set (i.e. sampled iid from the same distribution).  For each overall metric, we can compute an analogous standard error as the standard deviation of that metric on cluster-bootstrap resample of the test set [2] where on each bootstrap sample we draw full clusters from the test-set clusters with replacement. These standard errors approximate the variability in the overall metrics owing to the choice of structures included in the test set. We will include these standard errors in our revision.
>
> Fine-tuning on dG: we thank the reviewer for this suggestion. We have not yet explored predicting direct dG’s because we expect this task to introduce additional complications; for example, the log-likelihood initialization is biased by the length of a sequence such that longer sequences will be predicted to be more stable than shorter sequences.  However, we hope to explore this direction in future work.
>
> Boltzmann-Alignment and ProteinMPNN-DDG: we will include a discussion in the relevant works section.
>
> We hope that addressing these concerns helped strengthen our submission!
>
> [1] Dauparas et al. Robust deep learning-based protein sequence design using ProteinMPNN. Science.
>
> [2] Cameron, A. Colin and Miller, Douglas L. A Practitioner’s Guide to Cluster-Robust Inference. Journal of Human Resources.

---

> > ### Comment · Reviewer_aZe7 · 2025-04-02
> >
> > Thank you for answering my concerns. I think adding some discussion on these per-residue patterns would be worth including in the final version. I think the additional experiments with the other deep learning-based baselines on SKEMPI are also crucial for this work to be convincing. I would also like to point out that Boltzmann-Alignment just made their [code](https://github.com/aim-uofa/BA-DDG) available recently, and comparing with with them would also be beneficial.
> >
> > Overall, I think this is a strong work, and I have raised my score accordingly.

---

### Official Review · Reviewer_tCAx · 2025-03-12

**Overall Recommendation:** 3

**Summary:**

This paper proposes a novel approach to modeling binding energy by leveraging folding energy and fine-tuning a protein inverse folding model. The proposed STAB-DDG model demonstrates improved performance in predicting binding energy, an area that has often been lacking in experimental results. This method effectively utilizes folding energy data to model binding energy, resulting in better performance compared to baseline models.

**Claims And Evidence:**

Yes.

**Essential References Not Discussed:**

See comment part.

**Experimental Designs Or Analyses:**

The paper included necessary experimental comparisons with existing models while the metrics in this field is rather limited.

**Methods And Evaluation Criteria:**

Yes.

**Other Comments Or Suggestions:**

N/A

**Other Strengths And Weaknesses:**

1. The paper is well-written, with clear problem definitions.
2. The proposed model can be naturally generalized to double or multiple mutations; however, the authors stated that data on multiple mutations were discarded. I suggest that the authors consider including comparisons with multiple mutation sites.
3. There are additional models, such as MutateEverything (https://arxiv.org/pdf/2310.12979), that are used for predicting stability. Did the authors attempt to compare their model's performance with these models or evaluate its performance in predicting folding energy?
4. The model's performance appears to be less impressive than that of ThermoMPNN. What advantages does fine-tuning ProteinMPNN provide? Did the authors also consider using sequence-based models for predicting folding energy, as the problem is fundamentally a stability prediction task?

**Questions For Authors:**

N/A

**Relation To Broader Scientific Literature:**

N/A

**Theoretical Claims:**

Yes, the proposed methods proved essential equations in context, such as properties of proposed models.

---

> ### Author Rebuttal · Authors · 2025-04-01
>
> Thank you for your review. We appreciate you pointing out our novelty of using folding energy data for binding ddG prediction. We address your comments/questions below.
>
> Including multiple mutation sites: We have now evaluated folding ddG performance on multi-mutants in the megascale test set and report the results below.
> |              | Pearson | Spearman | RMSE |
> |--------------|---------|----------|------|
> | Single       | 0.73    | 0.71     | 0.74 |
> | Multiple     | 0.37    | 0.41     | 1.35 |
>
> We find that StaB-ddG performs worse on multi-mutants. We suspect that this is because of the limited number of multi-mutants in the training data. We will include these results in the appendix. We clarify that the results on SKEMPI in our submission include multiple mutations.
>
> Why ProteinMPNN for stability prediction: The reviewer observes that since the problem is fundamentally a stability prediction task, other models that can predict stability (e.g. ThermoMPNN and sequence models) could be used instead of ProteinMPNN.  This is indeed the case. In our submission we chose ProteinMPNN for
>
> 1. Simplicity of applying to complexes: ProteinMPNN accommodates multi-chain complexes natively, whereas the other methods described are implemented only for monomers.  Adapting these alternative methods would require heuristics such as adding a glycine linker or a residue gap that might negatively impact performance,
>
> 2. ProteinMPNN is light-weight.  Compared to ProteinMPNN, ThermoMPNN includes an additional transfer-learning module, and MutateEverything is built on either ESM2 or AF2 which have >10X more parameters and longer runtime compared to ProteinMPNN for a forward pass,
>
> 3. The thermodynamic properties of the resulting predictor.  Unlike StaB-ddG, a predictor that uses ThermoMPNN or MutateEverything would be inherently asymmetric and so would not satisfy properties 1 and 2 in our Proposition 1, and
>
> 4. Strong zero-shot performance: Inverse folding models (including ProteinMPNN) provide stronger zero-shot stability performance than sequence models, and so provide a stronger starting point for fine-tuning [1].
>
> In our revision we will make clear in the text the reasons for this design choice.  Additionally we will include an evaluation of StaB-ddG with ESM-IF model as the stability predictor in our revision to confirm our result that including folding stability data improves binding prediction is not specific to our choice of ProteinMPNN.
>
> [1] Notin et al. ProteinGym: large-scale benchmarks for protein fitness prediction and design.  Neurips 2023.

---

> > ### Comment · Reviewer_tCAx · 2025-04-05
> >
> > The model's performance on the multiple mutation prediction task is not very good, but given that the authors used a dataset with a limited number of multiple mutations, I find this reason acceptable. Considering the novelty of using folding predictions to estimate binding affinity, I have revised my score.

---

### Official Review · Reviewer_yFSX · 2025-03-15

**Overall Recommendation:** 2

**Summary:**

This paper presents StaB-DDG, a finetuning method for predicting mutational effects on protein binding. Specifically, it uses proteinMPNN, an inverse folding model, to calculate folding energy for a protein and binding energy a protein complex. It then finetunes proteinMPNN on experimental folding and binding DDG data so that the likelihood of proteinMPNN aligns with experimental binding/folding energy. It also includes a consistency training method to make sure the StaB-DDG satisfying symmetry and transitivity. The method is evaluated on standard SKEMPI benchmark and case study on TCR mimics.

**Claims And Evidence:**

The claims made in the submission is clear, but it can be more convincing if the authors include more baselines on the SKEMPI benchmark  as there are many papers in this area.

**Essential References Not Discussed:**

N/A

**Experimental Designs Or Analyses:**

Results on the SKEMPI benchmark will benefit from comparison to additional baselines. The comparison to existing baselines may not be fair because the proposed method is trained on additional folding energy data.

**Methods And Evaluation Criteria:**

The benchmark datasets make sense for the problem. The case study on TCR-mimics are particularly interesting. However, the authors should compare with more baselines on the SKEMPI benchmark, including DiffAffinity, Prompt-DDG, ProMIM, Surface-VQMAE, and Light-DDG.

**Other Comments Or Suggestions:**

All suggestions are included above

**Other Strengths And Weaknesses:**

This paper lacks technical innovation. It is a simple finetuning of ProteinMPNN model, with antisymmetry and path independence constraint.

**Questions For Authors:**

Can you upload the filtered and clustered SKEMPI dataset for review? The proposed data is substantially different from standard practice.

**Relation To Broader Scientific Literature:**

The key contribution of this paper is including additional training data from folding DDG experimental data. It showed that including folding DDG data is helpful for binding DDG prediction. This is a interesting finding

**Theoretical Claims:**

There is no theoretical claims

---

> ### Author Rebuttal · Authors · 2025-04-01
>
> We thank the reviewer for their constructive comments and appreciate that they find it interesting that StaB-ddG allows folding ddG data to improve binding ddG prediction. We hope addressing the comments has helped strengthen our submission.
>
> Baselines: We agree with your comment that the paper will be made more convincing by including more baselines. To address this we have re-trained DiffAffinity, Prompt-DDG, ProMIM, and VQ-MAE on our SKEMPI splits, and will include the results below in table 4:
>
> | Method         | PS Pearson      | PS Spearman    | PS RMSE       | Pearson | Spearman | RMSE | AUROC |
> |----------------|-----------------|----------------|---------------|--------------|---------------|-----------|-------------|
> | DiffAffinity   | 0.262 ± 0.039   | 0.247 ± 0.037  | 1.55 ± 0.13   | 0.309        | 0.326         | 1.88      | 0.64        |
> | Prompt-DDG     | 0.319 ± 0.045   | 0.267 ± 0.044  | 1.41 ± 0.12   | 0.331        | 0.353         | 1.81      | 0.57        |
> | ProMIM         | 0.191 ± 0.055   | 0.153 ± 0.052  | 1.57 ± 0.12   | 0.345        | 0.347         | 1.85      | 0.60        |
> | Surface-VQMAE  | 0.371 ± 0.044   | 0.357 ± 0.039  | 1.40 ± 0.10   | 0.445        | 0.446         | 1.59      | 0.67        |
> | StaB-ddG       | 0.473 ± 0.035   | 0.433 ± 0.037  | 1.52 ± 0.14   | 0.542        | 0.489         | 1.79      | 0.72        |
>
> PS: per structure
>
> We find these methods perform worse than both StaB-ddG and Stab-ddG zero-shot on 5 of the 7 metrics (all but RMSE). We suspect this better performance by StaB-ddG is due to its folding energy-based parameterization, which provides improved generalization. Out of distribution generalization is particularly important for our stricter data splitting.
>
> Filtered and clustered SKEMPI splits: Thank you for pointing out that these clusters were not specified in our submission. We include these below, with each cluster on a separate line:
>
> Train:
>
> 1AK4
>
> 1B2S 1B2U 1B3S 1BRS
>
> 1C4Z
>
> 1E50 1H9D
>
> 1F47
>
> 1FFW
>
> 1IAR
>
> 1KBH
>
> 1QAB
>
> 1YCS
>
> 2AW2
>
> 2B42
>
> 2C5D 2C5D 4RA0
>
> 1EMV 2WPT
>
> 2HRK
>
> 2J0T
>
> 2KSO
>
> 2O3B
>
> 2VN5
>
> 3BP8
>
> 3BT1
>
> 3EG5
>
> 3EQS 3EQY
>
> 3F1S
>
> 1ACB 1AHW 1BJ1 1CBW 1CHO 1CSE 1CZ8 1DQJ 1DVF 1EAW 1FC2 1FCC 1GC1 1JRH 1MHP 1MLC 1N8O 1N8Z 1NCA 1NMB 1PPF 1R0R 1SMF 1TM1 1UUZ 1VFB 1XGP 1XGQ 1XGR 1XGT 1XGU 1YQV 1YY9 2B2X 2BDN 2FTL 2NY7 2NYY 2NZ9 2SIC 3BDY 3BE1 3BN9 3BX1 3G6D 3HFM 3L5X 3MZW 3N85 3NGB 3NPS 3SE8 3SE9 3SGB 3W2D 4GXU 4JPK 4KRL 4NM8 5C6T
>
> 1A22 1BP3 3MZG
>
> 3Q8D
>
> 3SE3 3SE4
>
> 3SZK
>
> 3VR6
>
> 4HFK
>
> 2DVW 3AAA 4HRN
>
> 4J2L
>
> 4JEU
>
> 4K71
>
> 4OFY
>
> 4PWX
>
> 4RS1
>
> 1OHZ 4UYP 4UYQ 5M2O
>
> 4Y61
>
> 5CXB 5CYK
>
> 5E6P
>
> 5F4E
>
> 5K39
>
> Test:
>
> 1B41 1FSS 1MAH
>
> 1EFN 1GCQ
>
> 1C1Y 1GUA 1HE8 1K8R 1LFD 3KUD 4G0N 5TAR 5UFE 5XCO
>
> 1KTZ 1REW 2QJ9 2QJA 2QJB 3B4V 3BK3 3HH2 3SEK
>
> 1S1Q 1XD3 2OOB 3M62 3M63
>
> 1A4Y 1Z7X
>
> 4CPA
>
> 1JTD 1JTG 2G2U 3QHY
>
> 2PCB 2PCC
>
> 2AJF 3KBH
>
> 3S9D 3SE4
>
> 3WWN
>
> 4B0M
>
> 4CVW
>
> 4E6K
>
> 1AO7 1BD2 1JCK 1LP9 1MI5 1OGA 1SBB 2AK4 2BNR 2P5E 2PYE 3C60 3HG1 3QDG 3QDJ 3QIB 4FTV 4JFD 4JFE 4JFF 4L3E 4MNQ 4N8V 4OZG 4P23 4P5T 5E9D
>
> 4FZA 4NZW 4O27
>
> 3SF4 4WND
>
> 4X4M
>
> 4YFD 4YH7
>
> Our final version will include dataset filtering/splitting scripts.

---

### Decision · Program_Chairs · 2025-05-01

**Decision:**

Accept (poster)

**Comment:**

The paper presents a novel approach for predicting mutational effects on protein that leverages relationship with biding energy via the use of a pre-trained protein inverse folding model. Exeperiments demonstrate that the proposed approach is able to outperform state-of-the-art Rosetta-based predictor Flex ddG.
The proposed approach is sound and the results convincing. The authors have done a great job at addressing the reviewers comments and we urge them to incorporate the added discussion and results in their manuscript as these significantly improve the significance of the contributions.
In particular, please include the new results on additional baselines provided in response to Reviewers yFSX and aZe7 and if possible consider adding comparison against the Boltzmann-Aligned Inverse Folding Model (ICLR 2025).
It would also be important to include results on multi-mutant as you suggested noting the limited data size. Please also include the results regarding RMSE by mutant residue type for single mutants on SKEMPI.